# Factors shaping the abundance and diversity of the gut archaeome across the animal kingdom

Courtney M. Thomas[1,2], Elie Desmond-Le Quéméner [ID] [3], Simonetta Gribaldo[1] & Guillaume Borrel [ID] [1✉]

Archaea are common constituents of the gut microbiome of humans, ruminants, and termites but little is known about their diversity and abundance in other animals. Here, we analyse sequencing and quantification data of archaeal and bacterial 16S rRNA genes from 250 species of animals covering a large taxonomic spectrum. We detect the presence of archaea in 175 animal species belonging to invertebrates, fish, amphibians, birds, reptiles and mammals. We identify five dominant gut lineages, corresponding to *Methanobrevibacter*, *Methanosphaera*, *Methanocorpusculum*, *Methanimicrococcus* and "*Ca*. Methanomethylophila-ceae". Some archaeal clades, notably within *Methanobrevibacter*, are associated to certain hosts, suggesting specific adaptations. The non-methanogenic lineage *Nitrososphaeraceae* (*Thaumarchaeota*) is frequently present in animal samples, although at low abundance, but may have also adapted to the gut environment. Host phylogeny, diet type, fibre content, and intestinal tract physiology are major drivers of the diversity and abundance of the archaeome in mammals. The overall abundance of archaea is more influenced by these factors than that of bacteria. Methanogens reducing methyl-compounds with $H_2$ can represent an important fraction of the overall methanogens in many animals. Together with $CO_2$-reducing metha-nogens, they are influenced by diet and composition of gut bacteria. Our results provide key elements toward our understanding of the ecology of archaea in the gut, an emerging and important field of investigation.

[1] Institut Pasteur, Université Paris Cité, UMR CNRS6047, Unit Evolutionary Biology of the Microbial Cell, F-75015 Paris, France. [2] Sorbonne Université, Collège doctoral, F-75005 Paris, France. [3] INRAE, Univ Montpellier, LBE, Narbonne, France. ✉email: guillaume.borrel@pasteur.fr

The intestinal microbiota plays key roles in host health[1–8]. It is composed of bacteria, archaea, microbial eukaryotes, and viruses/phages. Several studies have unveiled features that influence the overall structure of the intestinal microbiota such as diet or the ability to fly[9–12]. However, most of these studies have only targeted the bacterial intestinal community. Host-associated archaeal methanogens produce a significant amount of methane gas in ruminants, which makes them ecologically and environmentally important[13], and in humans, archaea have been linked to various conditions of health and disease[5].

An early study addressed the distribution of intestinal methanogens in a wide variety of animals using methane gas detection[14]. This study detected methanogens in a wide range of animals and suggested that they were acquired early in animal evolution and completely lost in some lineages such as the Carnivora. However, methane measurement has several limitations, as it cannot detect non-methanogenic archaea or methanogenic populations with low concentrations in faeces, and does not provide taxonomic information on which archaea are present. Knowledge on diversity of archaea associated to the animal gut is very fragmented and limited to a few hosts. Most archaea-centric intestinal microbiome studies have indeed been conducted on a narrow group of animals: termites, humans, and ruminants[5,15–24]. Single studies based on different molecular and cultural approaches have also identified intestinal archaea in rats, hoatzin, pigs, seals, wallabies, kangaroos, iguanas, fish or horses[17,25–29]. Overall, these analyses reported that the most common methanogens in the gut are members of the *Methanobacteriales* and *Methanomassiliicoccales*[5,13,17,30,31] but little is known about the presence and distribution of other archaea, especially non-methanogenic lineages. For example, Thaumarchaeota have been detected in the gut of great apes and humans[18,32], but their presence in other animal is unknown. Finally, there is a lack of quantitative data on the abundance of archaea (and even bacteria) in the host microbiota. Overall, this

lack of information has hindered the identification of factors influencing the composition, diversity and abundance of archaea in the animal gut.

In this work, we analyse gut samples collected from 250 species, covering a broad spectrum of animal diversity. Using both sequencing and quantitative approaches, we investigate host-associated archaea in eight animal classes and identify the major gut archaeal lineages, as well as the dominant methane metabolisms. Based on these data and a meta-analysis of the environmental distribution of all archaeal sequences in the Silva database we also predict several events of adaptation to the gut in the Archaea. Finally, by using a wide range of metadata from the literature, we investigate the factors influencing the composition, diversity and abundance of the gut archaeome across the animal kingdom.

## Results and discussion

**Archaea are present in the gut microbiome throughout the animal kingdom.** We analysed samples from 250 species of animals ($n = 341$ samples) ranging from invertebrates to mammals – the majority of which, except for birds (Aves), fish (Actinopterygii) and gastropods, came from captive specimens (Supplementary Data 1). We used three approaches to characterize the archaeal community of these samples: i) quantitative PCR (qPCR) targeting total Archaea, total Bacteria, and five archaeal lineages known to be present in the animal intestine (*Methanobacteriales*, *Methanomassiliicoccales*, *Methanomicrobiales*, *Methanimicrococcus* and Thaumarchaeota), ii) 16S rRNA gene amplicon sequencing of the Archaea only and iii) of the entire microbial community. With Archaea-specific sequencing, we detected the presence of archaea in the gut microbiome of 175 species belonging to all eight animal classes investigated, including 14 orders of mammals (Fig. 1a; Supplementary Data 1).

Archaea were detected in a higher proportion of the species when using archaea-specific primers for qPCR (77%) or amplicon

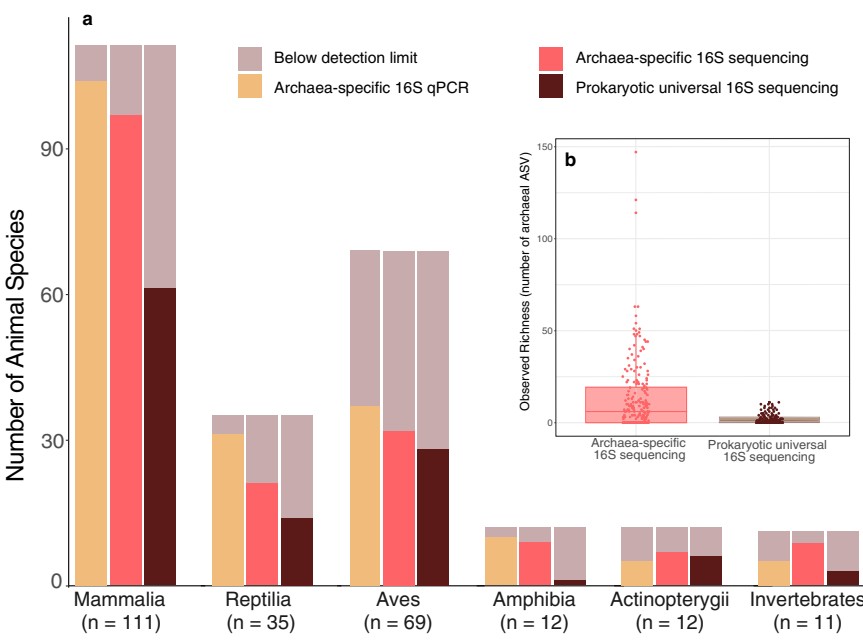

**Fig. 1 Detection of archaea in animal species with different approaches. a** Presence/absence using qPCR with archaea-specific primers, sequencing with archaea-specific primers and sequencing with prokaryotic universal primers. Invertebrates gather 3 classes (Insecta, Mollusca, and Malacostraca).
**b** Comparison of the number of archaeal ASV in sequencing data using archaea-specific and prokaryotic universal primers ($n = 218$ animal species). In the boxplots, the minima is minimum value, maxima is maximum value, center is median and quartiles are shown by box and whiskers, with individual animals shown as colored dots. In total, archaea-specific sequencing identified 1307 different archaeal ASVs while prokaryotic universal sequencing only identified 140 different archaeal ASVs.

sequencing (70%) with respect to the prokaryotic universal primers for amplicon sequencing (46%) (Kruskal-Wallis chi-squared = 56.928, df = 2, $p = 4.35e^{-13}$, $n = 250$). This difference was also observed in most animal classes (Fig. 1a). In addition, prokaryotic universal primers also captured a lower number of archaeal ASVs ($1.9 \pm 2.6$ ASVs per sample; 140 in total) with respect to the archaea-specific primers ($13.6 + 20.3$ ASVs per sample; 1307 in total) (Kruskal-Wallis $p = 1.65e^{-8}$ (Fig. 1b), $n = 218$). With ~10,000 prokaryotic reads (after rarefaction) per samples, which corresponds to a common number reads in recent studies relying on 16S rRNA gene sequencing[9,33,34], the archaeal species/ASVs that represent less than 0.01% of the microbial community were likely missed. This may explain why prokaryotic universal primers, although having no mismatches with most archaeal sequences, identify a lower number of archaea-positive animals and a lower archaeal alpha diversity than archaea-specific primers. Finally, the composition of the archaeal community was largely similar across the three approaches, especially for samples with more than $10^7$ archaeal 16S rRNA gene copies per gram of faeces (Supplementary Fig. 1).

**Dominance of five major unrelated lineages in the animal gut suggest independent adaptations.** The broad taxonomic coverage of the animal hosts and the use of archaea-specific primers allowed us to identify archaeal ASVs belonging to 19 described families, 10 orders, 6 classes, and 3 phyla. Despite this large diversity, the majority of the archaea present in the gut are closely related to cultured ones. Indeed, 84.9% of these ASVs (94.5% of the reads) share more than 95% identity with cultured species in the Living Tree Project (LTP, v138) database[35] amended with characterized candidate species, and half of the reads share more than 99% identity with known species (Supplementary Table 1). Consistently, the vast majority (93.7%) of the reads are affiliated to only six genera or families (Fig. 2a): *Methanobrevibacter*, *Methanosphaera* (*Methanobacteriales*), *Methanomethylophilaceae* (*Methanomassiliicoccales*), *Methanocorpusculum* (*Methanomicrobiales*), *Methanimimicroccus* (*Methanosarcinales*), *Nitrososphaeraceae* (*Nitrososphaerales*/Thaumarchaeota group 1.1b). These lineages also constitute more than 50% of the gut archaeome in 92% of the sampled animals and can be qualified as "dominant gut archaea". Among them, *Methanobrevibacter*, *Methanosphaera* and "*Ca.* Methanomethylophilaceae" had already been extensively reported in the gut microbiota of ruminants, humans and termites[5,17,21]. *Methanobrevibacter* members are by far the most dominant methanogens in our dataset – composing over a third of the total number of reads, followed by "*Ca.* Methanomethylophilaceae" members which accounted for 17.5% of the total reads (Fig. 2a). The other two most prevalent methanogen lineages, *Methanocorpusculum* and *Methanimimicroccus*, are reported less often in studies on the animal digestive tract[17,36,37].

The five dominant methanogen lineages in the gut are rarely reported in open environments, as revealed by a meta-analysis of the archaeal 16S rRNA gene sequences present in the Silva database, covering a broad range of environments (Fig. 2a). Moreover, these five lineages are often dominant in samples with the highest archaeal absolute abundance (Fig. 2b). In contrast, the rarest archaeal lineages (e.g., *Methanobacterium*, *Methanosarcina*, *Methanomassiliicoccaceae*) in the gut have been generally reported from non-gut environments such as sediments and wetland soils[31,38] (Fig. 2a). In addition, they are generally dominant in gut samples with the lowest archaeal absolute abundance (Fig. 2b), meaning that they only occur at low absolute abundance. Together, this suggests that members of these lineages have a lower capacity to develop in the gut than in non-gut

environments. Interestingly, several of the rare gut lineages are the closest phylogenetic relatives to the common/abundant gut lineages (Fig. 2a). These phylogenetically related rare and dominant gut genera/families belong to orders (*Methanosarcinales*, *Methanomicrobiales*, *Methanomassiliicoccales*, *Methanobacteriales*) that are also often present in digestors/bioreactors ("engineered" category on Fig. 2a), suggesting an early adaptation to high-resource availability in these lineages. These data suggest that some of the traits favoring development in the gut were already present in the last common ancestor of *Methanobacteriales*, *Methanomassiliicoccales* or *Methanosarcina + Methanimicroccus*, which may have facilitated the emergence of gut-specialized archaea in these lineages. Considering that the dominant gut methanogen lineages are rarely detected in open environments, and that closely related lineages are rarely present in the gut, a strong specialization to the gut microbiome likely occurred at the divergence of each of the five dominant gut methanogen lineages, suggesting at least five major events of adaptation to the gut in the Archaea (Fig. 2a).

A few other methanogen lineages may have also developed specific adaptations to certain gut conditions or specific hosts. For example, "*Ca.* Methanomassiliicoccus intestinalis" was detected in high prevalence/absolute abundance in the gut of elderly people with a disturbed gut microbiota and living in long-term residential care[20]. *Methanomicrobium mobile* was not detected in our samples and is almost never detected in open environments, but it can represent a large proportion of archaea in the rumen (Fig. 2a)[39], suggesting a high-level of specialization to this gut compartment.

**Non-methanogenic lineages are also present in the animal gut archaeome.** The family *Nitrososphaeraceae* (*Thaumarchaeota*), which gathers 15% of the total reads in our study (Fig. 2a), has rarely been reported in previous gut microbiomes studies (great apes and humans) and only when archaeal specific primers coupled with high-throughput sequencing (or nested PCR) were used[18,32,40]. Conversely, this lineage of ammonia oxidizers is common in terrestrial environments such as soils (Fig. 2a)[41–44]. The presence of these obligate aerobes in the gut is somehow surprising, but oxygen is available in some sections of the intestine and in proximity to the gut wall[45] - and some animals can host more aerobic communities than others[46]. Nevertheless, the combination of the sequencing and quantitative data reveals that *Nitrososphaeraceae*/*Thaumarchaeota* are generally present in low abundance in the gut (Fig. 2c), even in samples where they are the dominant archaea (Fig. 2b).

The three thaumarchaeal ASVs that gather the largest number of reads in our dataset (ASV4/ASV20/ASV21) are also widely distributed among animal species, ASV4 being the most widespread archaea in our samples (present in 65 animal species from 8 classes). Interestingly, these three ASVs correspond to the most prevalent and abundant archaeal phylotypes (named DSC1 and DSC2) among 146 soil samples from various biomes[44] (Fig. 2d). Because most animals live (eat, sleep, groom) on soil, one hypothesis might be that these dominant soil archaea are ingested by chance, which could explain why they are present in a wide range of animal at a low absolute abundance (Fig. 2b). However, a sequence closely related to ASV21 and DSC1 was previously found in the human gut[40] (Fig. 2d). Moreover, even though Thaumarchaeota group I.1c are among the dominant archaea in soil[44,47] and thus should also be commonly present in our samples if detected Thaumarchaeota are ingested by chance, we only identified two rare ASVs belonging to this lineage in one sample. Interestingly, ASV4/ASV20 and DSC2 are closely related to several "*Ca.* Nitrosocosmicus" species[43,48,49] (Fig. 2d). These

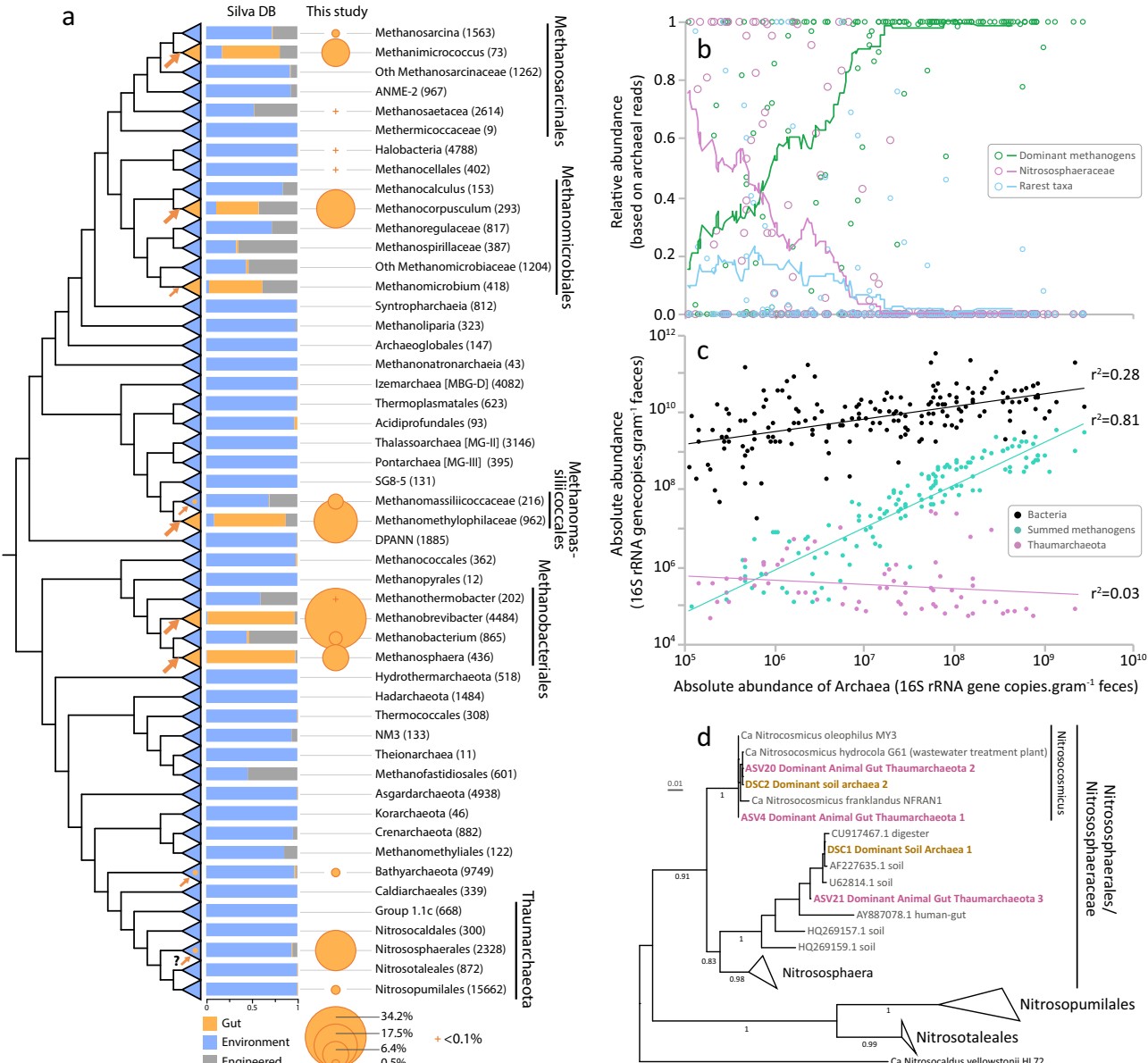

**Fig. 2 Main archaeal lineages in the gut and proposed independent events of adaptation to the gut in the domain Archaea. a** Distribution of archaeal 16S rRNA gene sequences in the gut and other environments based on sequences obtained from the Silva database and this study. The archaeal tree is based on Borrel et al.[112] enriched with DPANN lineages. Large arrows on the tree indicate five main events of adaptation to the gut environment, small arrows indicate four other possible events of adaptation to the gut. Fully orange triangles indicate that gut adaptation likely occurred at the base of the lineage while blue triangles with an orange spot indicate that gut adaptation occurred within the lineage. The histogram shows the proportion of sequences (from the Silva database) of a given lineage present in either animal digestive tract ("Gut", orange), open natural environment ("Environment", blue) or built environment ("Engineered", grey). Orange circle surface area represents the percentage of reads attributed to each taxon in our study (gut-related samples only). **b** Proportion of archaea corresponding to the dominant methanogen lineages (green), *Nitrososphaeraceae* (purple) and rarest taxa (light blue) in samples, based on amplicon sequencing (Miseq) with archaea-specific primers, according to absolute abundance of archaea in the sample (qPCR). Dots indicate the relative abundance of these three groups of archaea in each sample. Coloured lines indicate the moving averages of the relative abundance of these three groups on a subset size of 25 samples. The dominant methanogen lineages category contains *Methanobrevibacter*, *Methanosphaera*, "*Ca*. Methanomethylophilaceae", *Methanocorpusculum*, *Methanimicrococcus*. The rarest taxa category contains *Methanobacterium*, *Methanothermobacter*, *Methanomassiliicoccaceae*, *Methanosarcina*, *Methanoregulaceae*, *Methanospirillaceae*, *Methanosaeta*, *Methanocellales*, *Nitrosopumilaceae*, *Nitrosotaleaceae*, *Bathyarchaeota*, *Halobacteriales*. **c** Correlation between the absolute abundance (16S rRNA copies/gram of faeces) of archaea and bacteria (black), summed methanogen lineages (*Methanobacteriales*, *Methanomassiliicoccales*, *Methanomicrobiales*, *Methanimicrococcus*; green) and Thaumarchaeota (purple), all determined by qPCR using lineage-specific primers. Same samples (*n* = 176) are plotted in panels b) and c) and correspond to those with amplified archaea in deep sequencing (Miseq). The scale of the absolute abundance of archaea is the same than in **b**, **c**. **d** Phylogenetic position of dominant gut Thaumarchaeota (this study, ASV4, ASV20 and ASV21, purple, bold) and dominant soil archaea[44] (DSC1 and DSC2, brown, bold). ASV4/ASV20 are practically identical to DSC2 representative sequence (only 1 indel in a 4/5Gs homopolymer region, which may be due to a 454-sequencing error in DSC2[129]). ASV21 shares 99.2% identity with the DSC1 representative sequence.

**Table 1 Factors that influence the beta diversity of archaea in Mammals.**

| Beta diversity measure | Host Phylogeny (Mantel statstic) | Host order df = 10 | Gut diff Coeff n = 23 df = 1 | Diet df = 7 | GIT type df = 4 | Body mass df = 1 | qPCR archaeal abundance df = 1 | Arc:bac ratio (qpcr) df = 1 | Stomach pH n = 18 df = 1 | Mean retention time df = 1 | Origin* |
|---|---|---|---|---|---|---|---|---|---|---|---|
| Weighted unifrac | $r = 0.20$ $p = 0.003$ | $R^2 = 0.40$ $p = 0.001$ | 0.32 0.0009 | 0.20 0.002 | 0.10 0.02 | 0.06 0.002 | 0.04 0.01 | 0.04 0.03 | ns | ns | ns |
| Unweighted unifrac | $r = 0.34$ $p = 0.001$ | $R^2 = 0.33$ $p = 0.001$ | 0.17 0.0009 | 0.19 0.001 | 0.13 0.001 | 0.03 0.003 | ns | ns | ns | ns | 0.05 0.035 |
| Bray-Curtis | $r = 0.34$ $p = 0.001$ | $R^2 = 0.25$ $p = 0.001$ | 0.13 0.0009 | 0.16 0.001 | 0.14 0.001 | 0.02 0.01 | 0.02 0.03 | 0.02 0.01 | ns | ns | 0.05 0.018 |
| Jaccard | $r = 0.31$ $p = 0.001$ | $R^2 = 0.21$ $p = 0.001$ | 0.13 0.0009 | 0.14 0.001 | 0.11 0.001 | 0.02 0.02 | 0.02 0.02 | 0.02 0.02 | ns | ns | 0.05 0.018 |

Mammals with >2 species per order (n = 73, unless otherwise indicated) rarefied to 3000 reads per sample were subject to beta diversity analyses. *including only zoos from which more than three samples were collected, and samples from the same species were treated separately (n = 99; df = 11). The Mantel test was generated using a two-sided Pearson correlation method and used to determine the influence of host phylogeny on beta diversity across all metrics. Significant differences were tested for between beta diversity metrics using a permutational analysis of variance (PERMANOVA), p < 0.05 was considered significant. All PERMANOVA tests were one-sided. Gut diff coeff, coefficient of gut differentiation, which is the sizes of stomach and large intestine relative to the size of the small intestine (see ref. [88] and the text). GIT type, gastrointestinal tract type, which is the total morphology of the host intestinal tract -i.e., foregut fermenter, hindgut fermenter, and simple gut.

species can grow at ammonia concentrations (>20 mM) similar to those found in the gut[50–52], and that inhibit other ammonia-oxidizing Thaumarchaeota[43]. Further, one "Ca. Nitrosocosmicus" was isolated from a wastewater treatment plant[48], an environment that shares some characteristics with the gut. Therefore, an alternative hypothesis might be that the dominant Thaumarchaeota in the animal gut can remain in this environment, which may be beneficial for their dispersal in soils by animal faeces. The degree of adaptation and role of Nitrososphaeraceae in the animal gut remains to be elucidated.

Finally, although Bathyarchaeota were not common in our samples, most of the sequences we retrieved are closely related to a clade formed by "Ca. Termiticorpusculum" and "Ca. Termitimicrobium" (>95% id to termite sequences), two lineages recently identified in the termite gut[53]. Together with sequences from anaerobic digestors and sediments, our sequences from mammal, birds, reptiles, and crayfish, form a sister clade to termite sequences (Supplementary Fig. 2), suggesting that some general traits needed to be maintained in the gut are shared by these Bathyarchaeota.

**Specific associations between archaea and their hosts.** In mammals, the main factors that significantly affect the beta diversity of archaea are host taxonomy/phylogeny > coefficient of gut differentiation > host diet > digestive tract type (Table 1; Supplementary Fig. 3). There is a significant phylosymbiotic association between mammals and archaea (Mantel test, p = 0.001) (Table 1). The importance of host phylogeny/taxonomy and diet parallels what has been previously observed for the gut bacterial community of mammals[9,10,46,54–57]. Other factors such as the geographic origin of the samples and the body mass have little influence on the archaeal community structure (Supplementary text; Supplementary Fig. 4).

Specific associations between archaeal and animal host lineages are visible through the dominance of the gut archaeome by i) Methanobacteriales/Methanobrevibacter in Rodentia and most Cetartiodactyla, ii) Methanomassiliicoccales/Methanomethylophilaceae in Lemuridae, iii) Methanomicrobiales/Methanocorpusculum in Perissodactyla, and several reptiles or iv) Thaumarchaeota/Nitrososphaeraceae in Gastropoda (Fig. 3d). In addition, several archaeal clades are almost exclusively associated with a particular type of host. This is mainly the case for Methanobrevibacter which comprise clades associated with Primates, Cetartiodactyla, Perissodactyla and Rodentia (Fig. 4) suggesting specific adaptations to these hosts. These results are consistent with a previous report of Methanobrevibacter OTUs showing phylogenetic association with

hosts[10]. In Methanocorpusculum, Perissodactyla-associated clades are sisters to a Cetartiodactyla-associated clade (Supplementary Fig. 5), suggesting that the ancestor of these archaeal clades was already present in the ancestor of the Ungulata. Close relationships between Perissodactyla and Cetartiodactyla ASVs are also visible in Methanobrevibacter and Methanosphaera (Fig. 4). In "Ca. Methanomethylophilaceae", there is a large Primates-associated clade containing several of the typical human-associated species (Mx-03, Mx06[20]; Supplementary Fig. 6) for which the name of "Ca. Methanoprimaticola" has been proposed[58]. Fewer host-specific clades are observed outside mammals, except for reptile-specific clades in Methanocorpusculum (Supplementary Fig. 5). These clades complement the previously reported insect/termites-specific ones within Methanobrevibacter, Methanomethylophilaceae and Methanimicrococcus[21,36]) and support the hypothesis that archaea developed adaptations for specific host lineages with which they may have been associated for a long evolutionary time. In contrast, no clear host-associations are visible in Nitrososphaerales (Supplementary Fig. 7) which points to the absence of specialization to specific animal hosts and suggests a lower level of adaptation to the gut, as discussed above.

Host phylogeny also influences the absolute abundance and the alpha diversity of archaea. Indeed, mammals and reptiles tend to host higher abundances of archaea than other animal groups like amphibians, birds, fish, and invertebrates (Fig. 5a). This parallels what we observed for bacteria (Fig. 5b). In addition, Archaea were detected in more than 70% of the mammal, reptile, and amphibian species and less than 45% of the bird and fish species (based on archaea-specific amplifications, Fig. 1a; Supplementary Fig. 8). However, when archaeal lineages are considered separately, their absolute abundance is not significantly different between animal classes (Supplementary Fig. 9; Supplementary text). In Mammalia, samples from Perissodactyla, Cetartiodactyla, Primates, Diprotodontia and Rodentia species tend to have higher concentrations of archaea than other animals, whereas those belonging to Carnivora, Pholidota and Cingulata tend to have lower concentrations (Supplementary Fig. 10a). Conversely, the abundance of bacteria is more uniform across mammalian orders (Supplementary Fig. 10b). Closely related groups of animals also tend to have similar levels of archaeal alpha diversity, as supported by the Moran index (I = 0.08, p = 0.001, n = 150). For example, the archaeal richness is consistently high in the members of Gastropoda and in most members of the Cingulata, Equidae (order Perissodactyla) and Bovidae (order Cetartiodactyla) within Mammalia (Fig. 3b; Supplementary Data 1). Conversely,

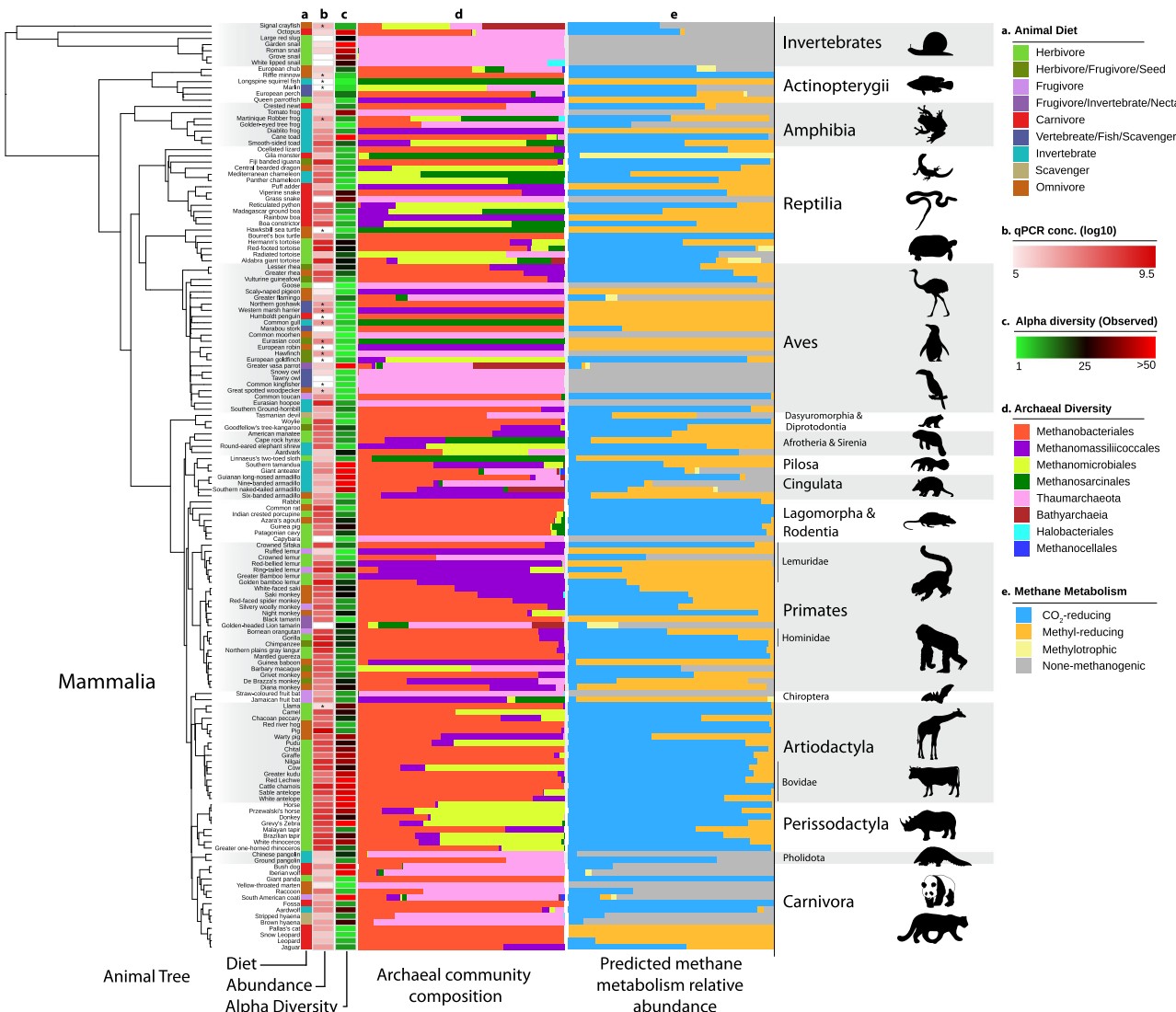

**Fig. 3 Archaeal abundance and diversity in the animal gut ($n = 150$; species with $\geq 3000$ archaeal reads). a** Information on animal primary diet gathered using the Elton Trait database, the Animal Diversity Website database, or from specialists who provided faecal samples. Primary diet was considered food material that made up $\geq 70\%$ of the animal's diet. **b** Absolute abundance of archaea as determined by qPCR with archaea-targeting primers on a log scale. Stars (*) indicate species for which the abundance may be underestimated (see Supplementary Fig. 19). **c** Observed richness (number of different ASV) of archaea. **d** Taxonomic diversity of archaea in the animal intestinal microbiome. Samples were rarefied to 3000 archaeal reads. e) Proportion of $CO_2$-reducing, methyl-reducing and methylotrophic methanogens, as well as non-methanogens in the archaeal community (see Supplementary Table 2 for assignation of metabolisms to taxa). The Animal Tree was generated using Timetree.org[103].

we found comparably low levels of archaeal richness within the Aves and Actinopterygii (Supplementary Fig. 11a).

**Strong influence of diet on methanogen abundance and composition**. Diet is another important factor affecting the gut archaeome in terms of alpha diversity, beta diversity (Table 1; Supplementary Fig. 11b) and abundance (Fig. 6). Indeed, herbivorous animals have a higher number of archaeal ASVs than carnivorous and omnivorous animals (Supplementary Fig. 11b). Moreover, the absolute and relative abundance of methanogens is higher in animals feeding on leaves than in animals feeding on meat or insects, and it tends to be intermediate in omnivorous animals (Fig. 6a). In contrast, ammonia-oxidising Thaumarchaeota are more abundant in carnivorous animals than in herbivorous animals (Supplementary Fig. 12a) which may be related to the higher availability of nitrogen compounds carnivor gut, and bacteria abundance did not show any significant differences

between diet types (Supplementary Fig. 12b). The link between methanogen abundance and diet type is further supported by the strong positive correlation of both the absolute and relative abundances of methanogens with the fibre content of the diet (Fig. 6b; Supplementary Fig. 13), contrasting with the absence of correlation for Thaumarchaeota and Bacteria (Supplementary Fig. 12c and d). The increase in methanogen absolute/relative abundance reaches a limit at around 200 g of crude fibre/kg of dry matter (Fig. 6b; Supplementary Fig. 13). At a lower host taxonomic level, the positive correlation also holds for Primates, for which we sampled species with contrasting average fibre intake (Fig. 6c). An increased fibre consumption was previously reported to be associated with a higher expression level of methanogenesis genes in humans[59] and greater methane production in pigs[60] and ruminants[61]. As the vast majority of intestinal methanogens are hydrogenotrophic, these relationships can be explained by the higher production of hydrogen from fibre/carbohydrates-rich diets (plant) than from protein/fat-rich diets (meat)[62].

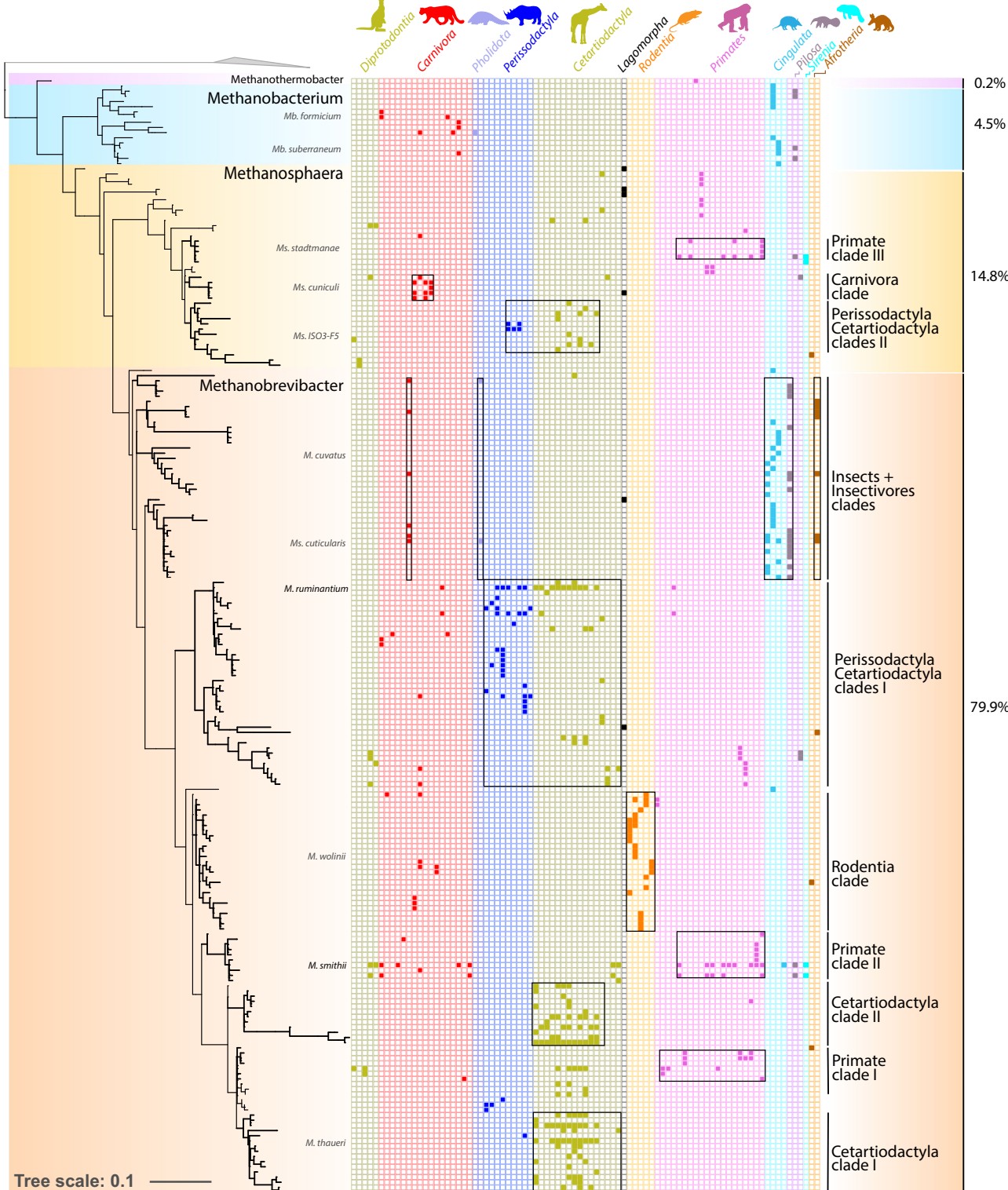

**Fig. 4 Distribution of *Methanobacteriales* ASVs among mammal species.** The phylogenetic tree (maximum-likelihood, GTR + G4) of *Methanobacteriales* was built with nearly full length 16S rRNA genes sequences from literature and the ASVs sequences from this study. For display purposes, the shown tree includes only the ASVs representing more than 1% of the sequences per sample, and no sequence from literature. Presence/absence of ASVs in animals is indicated by coloured highlighted/blank squares. Animal were gathered by mammalian orders, each with a different colour, as indicated on the top. Black boxes highlight archaeal clades preferentially present in a given host order, except for the Insect+insectivores clade composed of archaea preferentially present in insectivorous animals from different mammalian orders and insects (mostly termites, sequences from the literature). Clades corresponding to the boxes are labelled with thick lines on the phylogenetic tree. Species names in front of the tree indicate the position of cultured representatives of the Methanobacteriales. The percentages on the right indicate the proportion of reads from *Methanobacteriales* that were annotated as *Methanobrevibacter* (orange background), *Methanosphaera* (yellow background), *Methanobacterium* (blue background) and *Methanothermobacter* (pink background).

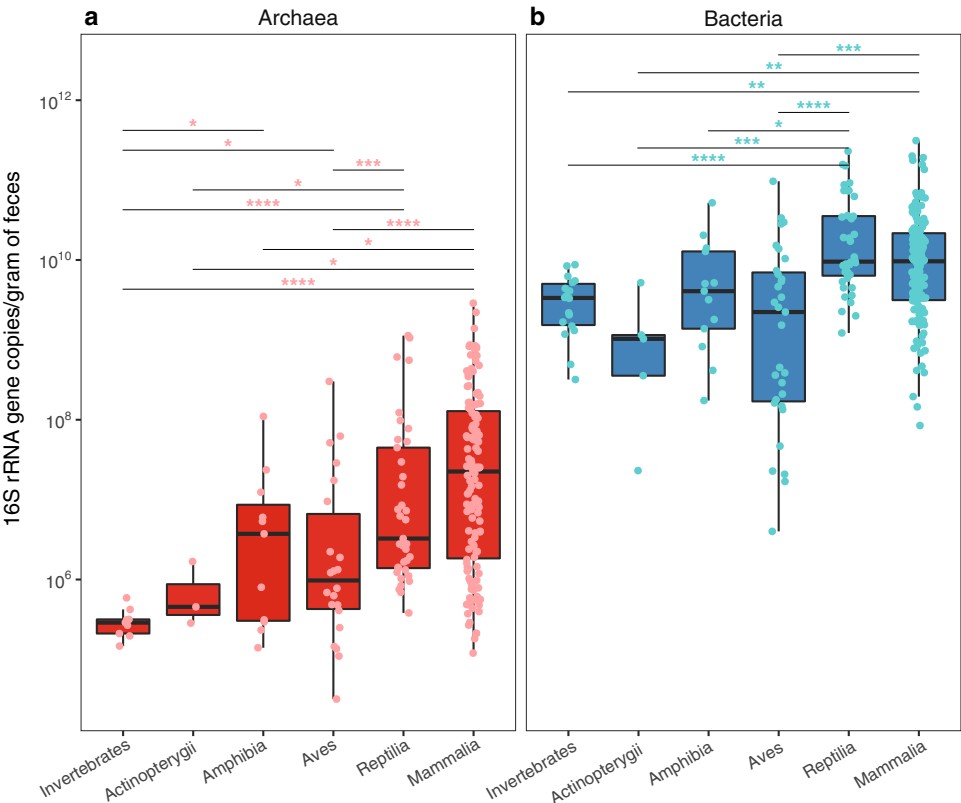

**Fig. 5 Absolute abundance of a) Archaea ($n = 221$, red) and b) Bacteria ($n = 255$, blue) determined by qPCR.** Animal lineages with significantly different archaeal/bacterial abundances are labeled. Two-sided Wilcoxon rank sum, *$p \leq 0.05$; **$p \leq 0.01$; ***$p \leq 0.001$; ****$p \leq 0.0001$. In the boxplots, the minima is minimum value, maxima is maximum value, center is median and quartiles are shown by box and whiskers, with individual animals shown as colored dots. Exact p-values are given in Supplementary Data 4.

However, the level of $H_2$ produced from fibre degradation also depends on which bacteria are involved. For example, several *Clostridiales* are known to produce more $H_2$ than *Bacteroides* during fibre degradation[63]. Thus, other than diet, methanogens should also be influenced by the composition of the bacteria degrading it. In humans, cellulolytic *Ruminococcaceae* (*Clostridiales, Firmicutes*) spp. have been reported to be present in the gut of methane producers, while cellulolytic *Bacteroides* spp. prevail in low/non-methane producers[63,64], and methanogens are enriched in subjects with the *Firmicutes/Ruminococcaceae* enterotype[65]. We found that eight *Ruminococcaceae* OTUs (including six from uncharacterized genera) co-occur with methanogens, and -more generally- 19 out of the 30 bacterial OTUs positively associated with methanogens belong to *Clostridiales* and only four to *Bacteroidales* (Supplementary Data 2, Supplementary text). Methanogen abundance is also correlated to bacterial abundance (Fig. 2c; Spearman correlation, $R^2 = 0.28$; $p = 10^{-14}$; $n = 176$) which may reflect the dependence of methanogens on bacterial metabolism. Other than benefiting from fibre degradation, methanogens can also favour it by stimulating microbes involved in its degradation. Indeed, the presence of methanogens in cocultures has been shown to increase the level of extracellular polysaccharide-degrading enzymes of *Ruminococcus flavefaciens*[66]. Interestingly, several *Christensenellaceae* OTUs, a bacterial family correlated with *Methanobrevibacter smithii* in human[67], are either positively or negatively correlated with *Methanobrevibacter* OTUs in ruminants (Supplementary Data 2), suggesting more complex interactions than previously thought.

The abundance of hydrogenotrophic methyl-reducing methanogen lineages (i.e., *Methanomassiliicoccales* and *Methanimicrococcus*)

is less influenced by fibre content than lineages that include hydrogenotrophic $CO_2$-reducing methanogens (i.e., *Methanobacteriales* and *Methanomicrobiales*; Supplementary Fig. 14). Moreover, hydrogenotrophic methyl-reducing methanogens represent a lower proportion of the methanogens in herbivorous animals than in animals having another type of diet (Two-sided T-test, $n = 115$, $p = 0.003$). As methyl-reducing methanogens depend on different methyl-compounds (e.g. methanol, methylamines) for their energy metabolism and because they can utilize hydrogen at lower concentration than $CO_2$-reducing methanogens[68], their distribution may be more affected by the availability of methyl-compounds than by fibre content. One of these methyl-compounds, methanol, is produced by the bacterial degradation of pectin[69]. This metabolism was shown to occur in the animal gut (e.g., human, pigs, lemurs, ruminants) as revealed by the identification of bacteria with a methylesterase activity[70,71] and by the increase in methanol concentrations in response to pectin consumption[72–74]. Our data show that the ratio of hydrogenotrophic methyl-reducing to $CO_2$-reducing methanogens is higher in frugivorous species than in herbivorous ones (Two-sided Wilcoxon test, $n = 51$, $p = 0.003$), which is likely related to large amounts of pectin in fruits. This support a previous hypothesis that the high relative abundance of *Methanosphaera stadtmanae* (an obligate methanol-reducing methanogen) in orangutan is related to their high fruit consumption[75].

We also found a high relative abundance of hydrogenotrophic methyl-reducing methanogens in most of the sampled Primates (Fig. 3e), and particularly in *Lemuridae*, which may be related to the presence of fruits in their diet (Supplementary Data 1). This relationship is further substantiated by the association between an archaeal OTU closely related to "*Ca*. Methanomethylophilaceae"

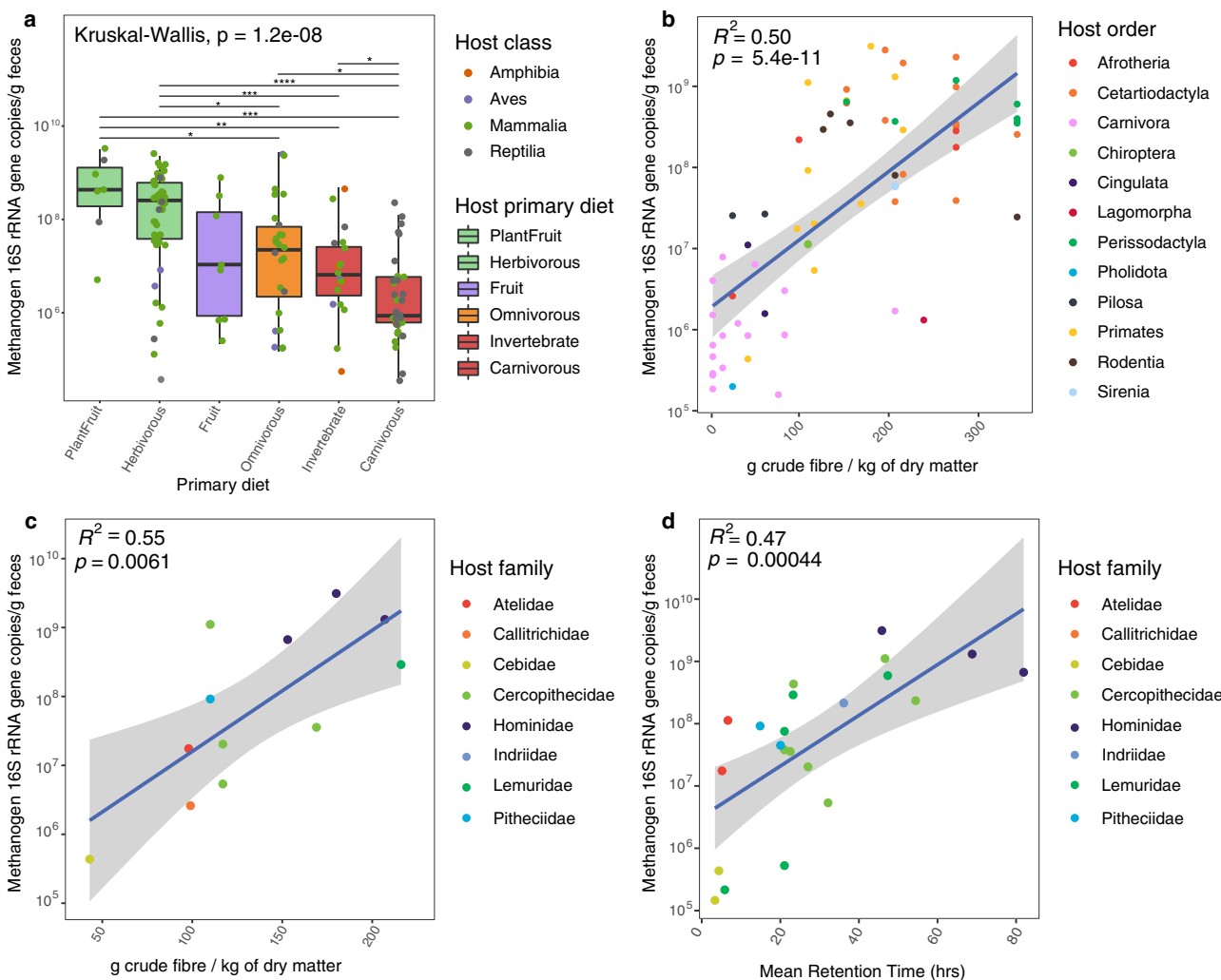

**Fig. 6 Influence of host diet type, diet-fibre content, and mean retention time on the absolute abundance of total methanogens. a** Abundance of total methanogens ($n = 139$) in animals grouped by diet type. The abundance of methanogens is the sum of individual quantifications of *Methanobacteriales*, *Methanomicrobiales*, *Methanomassiliicoccales* and *Methanimicrococcus* 16S rRNA genes. Two-sided Wilcoxon rank sum test with continuity correction was used to determine differences between diet types, $*p \leq 0.05$; $**p \leq 0.01$; $***p \leq 0.001$; $****p < 0.0001$. Significant differences across all groups were determined via the Kruskal-Wallis test, with $p < 0.05$ set as significant. Correlation between diet-fibre content and (**b**) absolute abundance of methanogens in mammal species ($n = 65$) and (**c**) in primates ($n = 12$). **d** Correlation between digesta mean retention time and averaged absolute abundance of methanogens in primates ($n = 22$). A two-sided, squared Pearson correlation coefficient was computed to assess the relationship between values, unadjusted $p$-values $< 0.05$ were considered significant. Grey bands around the lines (panels **b–d**) represent the 95% confidence interval around the linear regression model. Statistical analyses and representation of the absolute/relative abundance of methanogens were carried out on species where archaea have been detected. Exact $p$-values of panel (**a**) are given in Supplementary Data 4.

*sp.* Mx06[20] and a bacterial OTU closely related to *Lachnospira pectinoschiza* (OTUarc_11; OTUbac_2345; Supplementary Data 2). This bacterium grows mainly on pectin, producing methanol as a by-product of its degradation[76], and "*Ca.* Methanomethylophilaceae" sp. Mx06 has the genetic potential to grow by reducing methanol and methylamines with $H_2$[20]. A similar link may exist in humans, as both *Lachnospira pectinoschiza* and *Methanomassiliicoccales* abundance increases with age[77,78]. Moreover, "*Ca.* Methanomethylophilaceae" *sp.* Mx06 is the dominant archaeon in the gut of Yanomami Amerindians[20,79], whose diet is largely composed of fruits[80].

As *Methanomassiliicoccales* and *Methanimicrococcus* may also grow on other methyl-compounds than methanol, such as trimethylamine, they might be influenced by other types of diet. "*Ca.* Methanomethylophilaceae" OTUarc_11 is also correlated with an OTUs closely related to *Sarcina sp.* (OTUbac_4310; Supplementary Data 2) that can produce trimethylamine[81]. A

similar correlation between "*Ca.* Methanomethylophilaceae" and *Sarcina* was previously reported in the human gut[82]. Precursors of trimethylamine (i.e., glycine-betaine, carnitine and choline) are present in various diets[83–85] and pectin is not limited to fruit but is also a constituent of the plant cell wall[86], which therefore do not limit the presence of hydrogenotrophic methyl-reducing methanogens to frugivorous animals. In our dataset, hydrogenotrophic methyl-reducing methanogens constitute almost 40% of the overall methanogen reads (Fig. 7a; Supplementary Table 2) and represent a large fraction of the methanogens in many animals (Fig. 3e; Supplementary text). This contrasts with many non-host environments (e.g. sediments, peat bogs), where hydrogenotrophic methyl-reducing methanogens constitute a minor fraction of the overall methanogenic community[38,87]. It also reinforces the hypothesis that the gut environment is particularly propitious for this kind of methanogenesis, which could have led to the transition from methylotrophic (methyl-

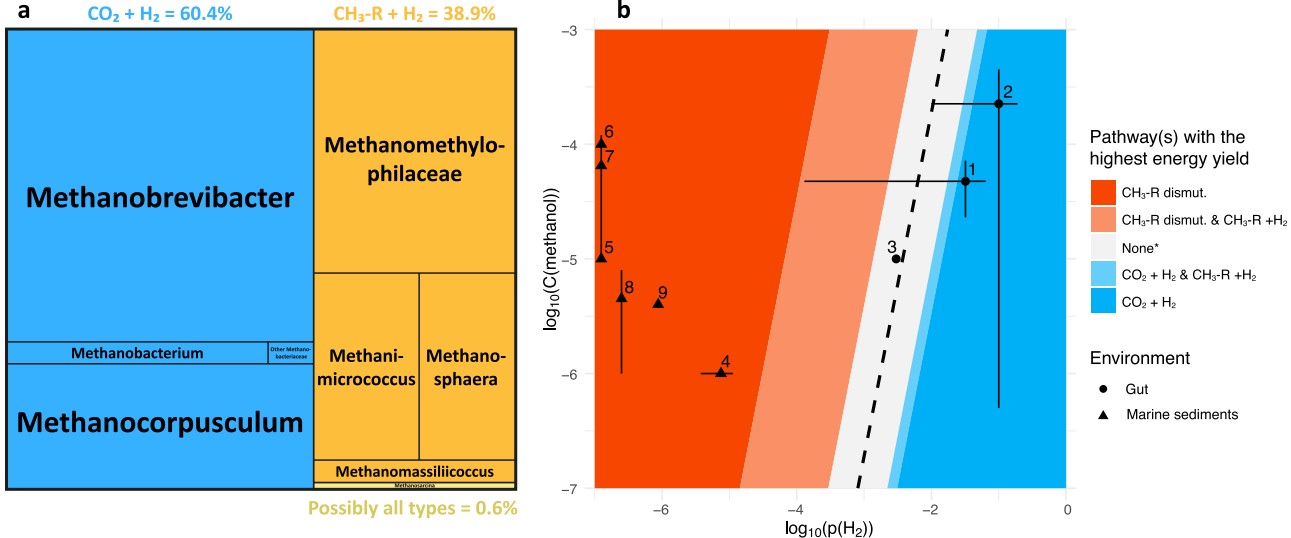

**Fig. 7 Main methanogenesis pathways in the animal gut. a** Proportion of the total archaeal reads that are assigned to taxa with a predicted hydrogenotrophic $CO_2$-reducing methanogenesis ($H_2 + CO_2$; blue) or hydrogenotrophic methyl-reducing methanogenesis ($CH_3$-R + $H_2$; orange) pathway. *Methanosarcina* spp. can have diverse methanogenesis pathways (i.e., the two above-mentioned pathways and the methyl-dismutation (or methylotrophic) and acetoclastic pathways). **b** Diagram indicating the methanogenesis pathways with the highest energy yield depending on methanol concentration (C(methanol) in mol/l) and hydrogen partial pressure (p($H_2$) in bar). The three methanogenesis pathways considered are methyl-compound dismutation ($CH_3$-R dismut.), hydrogenotrophic methyl-reducing methanogenesis ($CH_3$-R + $H_2$) or hydrogenotrophic $CO_2$-reducing methanogenesis ($CO_2 + H_2$). Coloured areas on the map indicate which pathway(s) yield(s) the highest amount of energy per mole of methane, i.e., concentrations and pressures for which the associated ΔG expressed in kJ/mol $CH_4$ is the lowest (see "Gibbs free energies of methanogenic pathways" in Materials and Methods). In central areas of the diagram, energy yields of two or three (*in this case none is yielding more energy) of the pathways are comparable with differences in ΔG of less than 10 kJ/mol $CH_4$. This is shown in light red, grey and light blue areas. The dotted line indicates values of C(methanol) and p($H_2$) for which all three catabolisms have exactly the same ΔG. Ranges of C(methanol) and p($H_2$) found in the literature for rumen (1), human colon (2) and cockroach hindgut (3), and marine sediments (4–8; Supplementary Table 3) were mapped on the graph: dots correspond to mean values and bars indicate minimal and maximal values.

dismutation) and $CO_2$-reducing methanogenesis to methyl-reducing methanogenesis in *Methanimicrococcus* and *Methanosphaera*, respectively[36]. Other methane metabolisms (based on dismutation of methyl-compounds or acetate) are almost absent from the animal gut and may occur in the few *Methanosarcina* members (0.6% of the total reads) identified in our dataset (Fig. 7a).

We explored whether energetic considerations could explain why methyl-compound dismutation is not found in the gut. For this, we compared the energy available through three different metabolisms (methyl-compound dismutation, hydrogenotrophic methyl-reduction, hydrogenotrophic $CO_2$-reduction), at different $H_2$ partial pressure (p($H_2$)) and methanol concentration (C(methanol)). Areas indicating pathways with the highest energy yields depending on the values of p($H_2$) and C(methanol) are mapped on Fig. 7b. For example, the red area on the left corresponds to low p($H_2$) values where methyl-dismutation gives more energy than hydrogenotrophic methyl-reduction or hydrogenotrophic $CO_2$-reduction per mole of methane. Using the few data available from literature, we compared conditions and dominant types of methanogenesis between marine sediments (triangles in Fig. 7b) and gut environments (circles in Fig. 7b). Values associated with marine sediments are found on the left in the zone where methyl-dismutation is the most favourable pathway, consistent with the dominance of this type of methanogenesis in this environment. Values associated with gut environments are found on the right in the zone where hydrogenotrophic metabolisms are the most favourable or as favourable as methyl-dismutation. Thus, energetic considerations only partially explain the almost complete absence methyl-dismutating methanogens in the gut, suggesting the existence of other unknown factors.

Finally, the influence of diet is further substantiated by presence of specific archaea in the gut microbiota of both predators and their preys. Indeed, within *Methanobrevibacter*, sequences of insectivorous mammals from distinct orders (Aardwolf in Carnivora, Tamandua in Pilosa, Ground pangolin in Pholidota, Armadillo in Cingulata, Aardvark in Afrotheria) are clustered with sequences from insect clades reported in the literature (Fig. 4)[21]. A similar phylogenetic clustering of insectivores and insect-derived archaeal sequences was not observed for other known insect clades outside of *Methanobrevibacter* (even if a few non-mammal insectivores are present in the *Methanimicrococcus* insect clade (Supplementary Fig. 15)). This suggests that some *Methanobrevibacter* species may develop in both insect and insectivorous mammal gut, while other insect-associated methanogens are mostly transients in the gut of insectivores. Interestingly, several *Thaumarchaeota* ASVs are shared between some Carnivora and Gastropoda (snail/slug) species (Supplementary Figs. 7 and S16). Whether this may reflect a diet link is unknown and requires further investigation.

**Impact of digestive tract physiology.** Both the coefficient of gut differentiation[88] (i.e., proportion of the gut dedicated to fermentation) and where the fermentation takes place (e.g. foregut, hindgut, caecum) explain part of the variance in the beta diversity (Table 1). In addition, many ASVs are almost ubiquitous in Cetartiodactyla having a rumen compartment (paraphyletic, Ruminantia and Tylopoda), but mostly absent from non-ruminant Cetartiodactyla or other animals, highlighting the possible dual influence of gut physiology and host-phylogeny (Supplementary Fig. 17). Whether these archaea found in faeces originate from the rumen compartment or can colonize more

largely the gut of these animal is currently unknown. The total abundance of methanogens is positively correlated with gut differentiation coefficient in mammals ($R^2 = 0.33$, $p = 0.0036$, $n = 25$; Supplementary Fig. 18a), while there was no correlation with abundance of Thaumarchaeota and Bacteria (Supplementary Fig. 18b, c).

It was previously reported that digesta mean retention time (MRT) is positively correlated with methane emission in herbivorous mammals[89] and positive relationships between methanogen abundance and MRT has been reported in humans[90]. When considering only Primates species, we identified a positive correlation between MRT and methanogen (or total archaea) abundance (Fig. 6d). However, we found only a weak positive correlation between MRT and methanogen (or total archaea) abundance in all animals (Pearson correlation, $R^2 = 0.16$; $p = 0.002$, $n = 59$). However, the distribution of the values suggests that the abundance of methanogens is mostly influenced by the lower range of MRT values. Indeed, there is a stronger positive correlation (Pearson correlation, $R^2 = 0.34$; $p = 1.5e\text{-}05$, $n = 47$) for MRT values ranging from 0.5 to 50 h and significantly less archaea in animals with an MRT < 20 h than animals with an MRT > 20 h (Wilcoxon rank sum test; $p = 4.4e\text{-}05$, $n = 65$). Diet type and MRT are generally related since digestion of a fibre-rich diet generally relies on microbial fermentation ("allo-enzymatic" digestion) which requires longer MRT than digestion of protein/soluble carbohydrate-rich diets that are processed at high rate by animal enzymes ("auto-enzymatic" digestion)[91,92]. An exception is the extreme case of the giant panda, an herbivore with a short MRT (8 h) that is a clear outlier in the relationships between fibre content and both methanogen absolute/relative abundance (Supplementary Fig. 13). The giant panda evolved from a carnivorous ancestor which may explain this short retention time (mammal carnivora have a low MRT). Conversely, carnivorous reptiles have a long MRT, which can exceed a week for some large snakes[93]. Among them, the boa constrictor and reticulated python have a high abundance of methanogens compared to other carnivorous animals which suggests that a long MRT can allow a substantial development of methanogens on meat diet. An increase in the relative abundance of *Firmicutes*, potential partners of methanogens, was also previously recorded in Burmese pythons during digestion[94]. However, while long MRT may facilitate methanogen development on a meat-based diet, it should be stressed that most carnivorous reptiles have a low abundance of methanogens. Outside of mammals and reptiles, most birds, fish, amphibians, and invertebrates have a low concentration of archaea (Fig. 5a). Many flying birds feeding on plant materials use only readily digestible components, and rapidly expel recalcitrant cell wall constituents without significant microbial fermentation[95]. This was suggested to be an adaptation to improve flight power by decreasing body mass[95]. It is thus likely that the short transit time and the low level of plant fibre fermentation have a negative impact on methanogen abundance in birds. In addition to low concentrations of methanogens in Carnivora, birds and fish, we found no clear archaeal clades associated with these animals (with few exceptions, i.e., a small Carnivora-associated clade in *Methanosphaera*; Fig. 4), suggesting that no lineage of methanogen developed strong adaptations to these hosts.

It has been proposed that some animals, including birds, rely relatively little on their gut microbiota[96]. In addition, bacteria recovered from birds show little host specificity and do not display phylosymbiotic patterns with their host or correlation with diet, differently from what has been generally observed in most mammals gut microbiota[9]. In our dataset, the low abundance of bacteria in the gut microbiota of bird supports the hypothesis of Hammer et al.[96] and extend the observations of

Song et al.[9] on the particularity of the gut microbiota of these animals. However, we found that concentrations of faecal bacteria in other animals proposed to rely less on their gut microbiota, such as Carnivora species[96], are as high as in other mammals (Supplementary Fig. 10b).

**Comparison with a large-scale study on the animal gut archaeome.** An analysis of the diversity of the gut archaeome in a wide range of animals was published late 2021[97]. This study investigated samples coming largely from wild vertebrates while our study also included invertebrates and a larger proportion of captive animals. While both studies utilized amplicon sequencing to characterize the intestinal archaeome, we also acquired quantitative data (qPCR) for all main groups of gut archaea, as well as for whole archaea and bacteria.

Only 37 animal species are in common between both studies, among which 23 have archaea detected in both cases. Despite this low overlap in the taxonomic sampling, several results are congruent between the two studies. In both analyses, archaea were detected in a wide range of animals (110 species representing 69% of all species in Youngblut et al.[97], and 175 species representing 70% of all species in our study). Collectively, the two studies detected archaea in 262 animal species, out of 372 analysed animal species (Supplementary Data 3). Both studies also found that *Methanobacteriales* are the most dominant methanogens in many species, that variations in archaeal composition are low between individuals belonging to the same animal species, and that a majority of the ASVs have <97% sequence identity with currently isolated archaeal species. Regarding the latter point, our study also reports that this proportion dropped when archaea with a sequenced genome and highly enriched in culture (mostly *Methanomassiliicoccales* and *Nitrososphaerales*) were added to the comparison (Supplementary Table 1). The archaeal lineages identified in both studies are largely consistent, with the exception of *Methanothermobacter*. Youngblut et al.[97] found this genus in birds and linked it to species with higher body temperature. In contrast, despite being successfully targeted by our primers, *Methanothermobacter* was rarely detected in our samples, including those from wild birds. Like Youngblut et al.[97], we identified sequences of Bathyarchaeota (a typical sediment-associated lineage) in several animal species. In addition, our study shows that gut *Bathyarchaeota* are phylogenetically closely related to each other and form a clade sister to *Bathyarchaeota* previously identified in the insect gut (Supplementary Fig. 2)[53]. We also highlighted the high prevalence of *Thaumarchaeota*, a lineage rarely detected in the gut environment previously (Fig. 2). Beyond archaeal taxonomy, our study also predicted the main methanogenesis pathways in the gut environment and their distribution among animals (Fig. 3).

Both studies revealed the greater influence of host phylogeny than diet type on archaeal diversity. Our study also found an influence of these factors (and a few others like diet fibre content and digesta mean retention time) on archaea/methanogen absolute abundance (Figs. 5 and 6), and linked animal diet with the distribution of main methanogenesis pathways. Interestingly, Youngblut et al.[97] found that archaea have the strongest signal of co-phylogeny with herbivorous mammals, an aspect that we did not investigate. A little influence of phylogeny and diet on archaeal alpha-diversity was found by both studies, although we identified significant differences in archaeal richness between diet types and between host groups (Supplementary Fig. 11). Youngblut et al.[97] identified several ASVs whose relative abundance is specifically influenced by host phylogeny and diet. In agreement with these results, we identified archaeal clades associated with

specific host orders, mainly in *Methanobacteriales* (Fig. 4) and *Methanocorpusculum* (Supplementary Fig. 5). In particular, we found that Cetartiodactyla having a rumen compartment share a large core of *Methanobacteriales* ASVs that are unique to them (Supplementary Fig. 17). Little connections between the distribution of archaea and bacteria were found in both studies. In our case, we found that methanogens predominantly associate with *Clostridiales* species (Supplementary data 2), which may produce larger amounts of substrates for methanogenesis and developed association with them on a long evolutionary period. Both studies predicted ancestral states/events relative to host colonization by archaea. Youngblut et al.[97] predicted that *Methanobacteriales* members were present in the last common ancestor of all mammals. In our study, we identified five main transition events, from open environments to the gut environment, in the Archaea domain (Fig. 2a).

Altogether, our work provides key insights into the lifestyle and role of intestinal archaea across a diverse range of animals. We identified several events of adaptation to the gut which will be the base for future investigation of specific traits associated with gut-colonization in the archaea. Similar to what was previously reported for bacteria, the diversity and composition of archaea are influenced by host phylogeny and diet. In addition, we found that the abundance of archaea, mostly composed of methanogens, is also strongly influenced by these factors while bacterial abundance is more homogeneous. Increased sampling efforts, time-series analyses, and metagenomic investigation will help to answer standing questions about the impact of geography, captivity, residency, and adaptations of intestinal archaea throughout the animal phylogeny.

## Methods

**Sample collection and DNA extraction.** A majority of animal faecal samples were donated from various zoological institutions in France (Supplementary Data 1). After consultation with the corresponding Ethics Committee in Animal Experimentation our study got direct clearance since the samples that were used for this project were collected either from species or by methods that are not in the scope of the current regulation protecting the animals used for scientific purposes (European Directive 2010/63/EU, and French Decree 2013-118). Fresh faecal samples ($n = 391$) from 269 species were stored at $-20\,°C$ until DNA extraction. Total DNA was extracted using a modified QIAamp PowerFecal DNA Kit (Hilden, Germany) protocol. Cells were lysed using the Fastprep (MP Biomedicals) cell homogenizer 'faecal sample' default setting in the lysis buffer provided in the PowerFecal DNA kit. For subsequent analyses, genomic DNA was diluted ten times, to limit the effect of PCR inhibitors.

**Quantitative PCR.** Total bacteria, total archaea, and specific archaeal lineages (*Methanobacteriales*, *Methanomassiliicoccales*, *Methanomicrobiales*, *Methanimicrococcus*, Thaumarchaeota) were quantified using quantitative PCR with lineage specific primers (Supplementary Table 4). qPCR was performed on a qTower3 Touch device (Analytik Jena GmbH) using SensiFAST SYBR® & Fluorescein Kit (Bioline, Paris, France). For each run, a standard curve was prepared using a 10-fold serial dilution ($10^9$ to $10^1$ copies/μl) of a plasmid containing a 16S rRNA. Plasmids containing a partial archaeal or bacterial 16S rRNA gene were generated through cloning PCR amplified 16S rRNA gene of the groups into *E. coli*. Bacterial 16S rRNA genes were amplified from a faeces sample using the B-27F-YM/B-1492R primer set[98,99]. Archaeal 16S rRNA genes were amplified from *Methanimicrococcus blatticola*, *Methanocorpusculum aggregans*, "*Ca.* Methanomethylophilus alvus", *Methanosphaera stadtmanae* and *Nitrososphaera viennensis* using the A-21F/A-1386R primer set[100,101]. PCR products were cloned with a pGEM-T vector according to the manufacturer's instructions (Promega, Charbonnières-les-Bains, France). The accuracy of the plasmid construction was confirmed through sequencing and all plasmids were diluted to $10^9$ copies/μl, aliquoted and stored at $-20\,°C$. The accuracy of the qPCR assay was confirmed through melting curve analysis. All quantifications were performed twice in independent runs. The final concentration of all the microbial was averaged between replicates and normalized as copies of 16S rRNA gene per gram of faeces. 50 samples from 19 species, with no/low DNA detected after extraction, also had no amplification with any primer set (including the bacterial-specific primers) and were removed from subsequent analyses, leaving 341 samples from 250 species. In addition, several samples with a low 16S rRNA gene copy number to DNA concentration ratio, mostly corresponding to Aves and Actinopterygii species, were identified (Supplementary Fig. 19). As this low ratio may be indicative of PCR assay of inhibition or high

amount of host DNA, these samples were removed from subsequent analyses of qPCR results.

**16S rRNA gene amplicon sequencing.** Archaeal 16S rRNA genes were amplified in two steps (Nested-PCR; Supplementary Table 5) to allow the inclusion of a larger range of samples. Prokaryotic 16S rRNA genes were directly amplified with Illumina tagged primer pairs (Supplementary Table 5). Sequencing was performed on an Illumina MiSeq platform (Biofidal, Vaulx-en-Velin, France) according to the Illumina protocols for PE 2 × 300 bp, and resulted in more than 21 million reads and more than 16.7 million reads for the prokaryotic and archaea specific sequencing, respectively.

**Microbial Diversity Analyses.** Reads were processed and assigned to amplicon sequence variants (ASVs) using the DADA2 software (v1.12.1) in R (v3.6.0). Briefly, reads were trimmed and quality-filtered using the standard parameters - maximum expected errors for forward and reverse reads = 2, quality score = 2, and trimming length = 273 and 170 base pairs for forward and reverse reads, respectively. Forward and reverse reads were merged with a 20 base pair overlap, ASVs were generated, and chimeras were discarded. ASV annotation was performed using the Silva 16S rRNA database (v132). Assignment of ASVs to a main type of methane metabolism (hydrogenotrophic $CO_2$-reducing, hydrogenotrophic $CH_3$-reducing, acetoclastic and methylotrophic (methyl-dismutation)), was done based on their taxonomic affiliation, since all members of almost all methanogen genera/families have the same dominant type of methane metabolism (Supplementary Table 2). *Methanosarcina* is the main exception, as species from this group can have one or several types of methane metabolisms. All ASVs that were not annotated as archaea were removed from the archaea-specific primer generated sequences, and ASVs annotated as archaea or bacteria were kept from the prokaryotic universal primer generated sequences. Samples from the same species were merged by summing ASV abundances. These approaches resulted in 1307 archaeal ASVs from the archaea specific primers, as well as 140 archaeal ASVs and 19,145 bacterial ASVs from the prokaryotic universal primers. To estimate the novelty of the archaeal ASVs (obtained with the archaea-specific primers), we compared them using BLAST to 16S rRNA genes of isolated archaea retrieved from the Silva Living Tree Project LTP database (ltp_12_2020)[102] plus additional sequences of candidate species belonging to *Methanomassiliicoccales* and Thaumarchaeota. For diversity analyses, rarefaction was performed to normalize sequencing depth to 3,000 reads, leading to 1,253 archaeal ASVs. Bacterial ASVs were normalized to a sequencing depth of 12,000 reads per sample. Observed richness (alpha diversity) was estimated and all beta diversity analyses were performed using the 'phyloseq' package in R (v1.30.0). Subsequent statistical analyses were performed using the base Rstudio 'stats' package (v3.6.0) as well as the R package 'vegan' (v2.5-6).

To test for significant differences using the various beta diversity metrics (Table 1) a permutational multivariate analysis of variation (PERMANOVA) from the R package 'vegan' (function adonis) was used. The Mantel Test from the R package 'vegan' (function mantel) and 999 permutations were used to determine the influence of the mammalian phylogeny on beta diversity metrics. The mammalian tree ($n = 73$) was generated using Timetree.org[103] and then converted to a distance matrix using the 'ape' (v5.6.1, function cophenetic.phylo) package[104] in R. A pairwise Wilcoxon rank sum test with continuity correction from the R package 'stats' (function pairwise.wilcox.test) was used to determine differences between the absolute abundance of archaea and bacteria in animal diet types, as well as between animal classes. Linear regressions from the R package 'stats' (function lm) were used to determine the relationships between the abundance (log-transformed) of methanogens, Thaumarchaeota and bacteria, and mean retention time (MRT) and dietary fibre consumption. Significance cut-off was $p < 0.05$ for all analyses. Type I errors were corrected using the Benjamini–Hochberg (BH) approach in all pairwise comparisons.

**Placement of ASVs within Reference 16S rRNA gene trees.** All archaeal ASVs were filtered on a per sample basis, to keep only ASVs representing at least 1% of the total number of reads of the sample. Reference sequences >1200 bp with a quality >95% were obtained from the Silva SSU 138 database[35], RDP database (11.5)[105], and an in-house dataset. Redundancy was removed from reference sequences with a 98% or 97% sequence identity threshold using the VSEARCH software[106]. For each archaeal order, long reference sequences were combined with the ASV sequences and were aligned using the G-IN-SI algorithm in MAFFT (v7.273)[107]. Phylogenetic trees were generated using the GTR + G4 + I model in the IQTREE (1.6.12) software[108]. For each animal, the ASVs representing more than 1% of the reads were mapped in front of the archaeal tree using ITOL[109]. Reference sequences were ultimately removed from the tree to only keep the ASVs sequences.

**Co-occurrence of archaea and bacteria.** To identify co-occurrence signal between archaea and bacteria across Mammalia, Reptilia, and Aves, we integrated the sequences from both the universal and archaea specific 16S rRNA gene amplicon sequencing. Only bacterial reads were selected from the universal 16S rRNA gene amplicon sequencing for this analysis. We used VSEARCH (v2.17.1)[106] to cluster ASVs into OTUs at 97% in order to reduce the size of the dataset and to filter out

truly low abundance lineages of microbes. Then, to merge these datasets in a way that accurately represented the microbial community in terms of relative abundance between archaea and bacteria, we normalized the two datasets both in terms of sequence depth and in terms of archaea-bacterial ratios -information which was gathered through qPCR data. OTUs that were present in less than 10% of the animal classes – Mammalia, Aves, and Reptilia independently- were removed. Following this, we implemented both the SPIEC-EASI (Spiec.Easi package v1.1.0,[110]) and the SparCC algorithms[111] (part of the Spiec.Easi package (v1.1.0)) in Rstudio (v3.6.0) to determine co-occurrence trends between archaea and bacteria. Networks were calculated with 1000 iterations. The output from these analyses were filtered using a 0.5 minimum threshold of edge stability (SPIEC-EASI) (Supplementary Data 2) and a $p$-value < 0.05 (SparCC), independently. Only the co-occurrence patterns identified by both algorithms were further analysed.

**Investigation of archaea distribution in the gut and other environments**. The 16S rRNA gene sequences from the Silva database[102] originate from thousands of studies covering all types of environments and have a robust taxonomic annotation integrating most of the phylogenetic diversity. All archaeal 16S rRNA gene sequences from Silva SSU 138 database longer than 800 bp and with more than 80% sequence quality, alignment quality and pintail quality were downloaded. Sequences from shotgun-sequencing metagenomes were removed because their environmental origin was not clearly indicated. The annotation of each sequence was retrieved from GenBank and used to classify them as "Gut", "Environment" or "Engineered" origin. Sequences from sponge, animal environments (e.g., nest) or polluted sites (e.g., dump) were not included. The relative abundance of each category was mapped on a tree of archaea built with genomic sequences used in Borrel et al.[112] as well as additional DPANN sequences not present in this study.

**Gibbs free energies of methanogenic pathways**. The following chemical reactions were considered for the calculation of Gibbs free energy changes (ΔG) associated with methanogenic pathways:

Methyl-compound dismutation (CH$_3$-R dismut.):

(1) $4/3 \text{ methanol} \rightarrow CH_4 + 1/3\ CO_2 + 2/3\ H_2O$

Hydrogenotrophic methyl-reducing methanogenesis (CH$_3$-R + H$_2$):

(2) $\text{methanol} + H_2 \rightarrow CH_4 + H_2O$

Hydrogenotrophic CO$_2$-reducing methanogenesis (CO$_2$ + H$_2$):

(3) $CO_2 + 4\ H_2 \rightarrow CH_4 + 2\ H_2O$

For each pathway, ΔG calculations were performed using the R package CHNOSZ[113] considering C(methanol) between $10^{-3}$ and $10^{-7}$ mol/l, p(H$_2$) between 1 and $10^{-7}$ bar, T = 298 K, pH = 7 and p(CO$_2$) = p(CH$_4$) = $10^{-1}$ bar. These calculations were used to identify pathways with the highest energy yields (the lowest ΔG) per mole of methane as a function of C(methanol) and p(H2). See also[45] for another representation of the ΔG associated with these pathways as a function of p(H$_2$).

**Origin of the metadata**. Information of animal species were collected from various literature sources and online databases. Diet information for mammals and birds were downloaded from the EltonTraits database[114], and information for other animal diets were annotated using the Animal Diversity Web database (Museum of Zoology, University of Michigan, https://animaldiversity.org/). Information on body weight were also gathered on this website. Information about coefficients of gut differentiation, pH, diet fibre content, and intestinal tract structure and mean retention time were gather from[3,88,115–119]. Information on methanol, H$_2$ CO$_2$ and CH$_4$ in the gut and in marine sediments originate from[20,45,120–128].

**Statistics and reproducibility**. No statistical method was used to predetermine sample size. No data were excluded from the analyses, except for samples for which the DNA extraction wasn't successful and led to no amplification (50 samples corresponding to 19 species), and those with low bacterial 16S rRNA gene concentration compared to the DNA concentration, removed for qPCR analyses (see Supplementary Fig. 19). The experiments were not randomized. The investigators were not blinded to allocation during experiments and outcome assessment. No statistical methods were used to predetermine sample sizes. Data met the assumptions of normality for instances in which parametric tests were used.

**Reporting summary**. Further information on research design is available in the Nature Research Reporting Summary linked to this article.

## Data availability
The 16S rRNA gene sequencing data generated in this study have been deposited in the GenBank database under the BioProject PRJNA810306. The 16S rRNA gene sequence data used in this study are available in the SILVA Living Tree Project LTP (ltp_12_2020)[102] https://imedea.uib-csic.es/mmg/ltp/ltp-2020/; Silva SSU 138 database[105] https://www.arb-silva.de/documentation/release-138/; RDP database (11.5)[105] http://rdp.cme.msu.edu/index.jsp. Metadata on animal species used in this

study are available in EltonTraits database[114]; Animal Diversity Web https://animaldiversity.org/.

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

## Acknowledgements

C.T. is supported by a PhD fellowship from Paris Sorbonne Université Science and by funds from the doctoral school Bio Sorbonne Paris Cité "BioSPC". G.B. acknowledges funding from the French National Agency for Research Grant Methevol (ANR-19-CE02-0005-01) and S.G. acknowledges Archaevol (ANR-16-CE02-0005-01). This study has received funding from the French Government's Investissement d'Avenir program, Laboratoire d'Excellence "Integrative Biology of Emerging Infectious Diseases" (grant n°ANR-10-LABX-62-IBEID). We thank the two anonymous reviewers for valuable comments. We thank the computational and storage services (TARS cluster) provided by the IT department, in particular Sophie Creno, at Institut Pasteur, Paris. We thank Najwa Taib and Pierre Garcia for their help with python and R scripts. We also thank all people who generously collected and provided samples (or made it possible to do so) for this study: Jérôme Fuchs from the Museum National d'Histoire Naturelle (MNHN), Frédéric Delsuc (Institut des Sciences de l'Evolution, UMR5554), Alexis Lécu and Olivier Marquis from Parc Zoologique de Paris, Baptiste Mulot and Hanae Pouillevet from Zoo de Beauval, Paola Dvihally and Dominique Gitton from Parc des Mammelles, Antoine Talarmin, Yann Reynaud, Stéphanie Guyomard, Matthieu Pot, Gaëlle Gruel, Séverine Ferdinand from Institut Pasteur de Guadeloupe, Thomas Godoc from Aquarium de Guadeloupe, Patrick Buisson and Sabine Collin at Palais de la Découverte, Jérôme Contignac and David Luis Garcias Warner from Nabau Projects/Grupo Atrox, Aude Bourgeois from Jardin des Plantes de Paris, Jérémy Sauvanet, Gaspard, Anaïs Tibi and Johnatan Aparicio. We thank Christa Schleper for providing a plasmid with a Thaumarchaeota 16S rRNA gene for qPCR standard, Jérôme Mathieu for data on gastro-intestinal tract types and Marcus Clauss for helping us collecting the MRT data from an abundant literature.

## Author contributions

C.M.T., G.B. and S.G. conceived the study. C.M.T. did the experiments and E.D. the thermodynamic calculations. C.M.T. and G.B. analyzed the data and wrote the manuscript with the input of S.G.

## Competing interests

The authors declare no competing interests.
