## [Peer Review File · Nature Communications]

REVIEWER COMMENTS

Reviewer #1 (Remarks to the Author):

This manuscript presents some interesting data on the characterisation of the archaeome in a variety of animals. Some of the findings on the link between archaeal diversity and host phylogeny (Fig 3) or correlation of methanogen abundance with diet-type over an extensive range of animals (Fig 6d) represent a substantial advance for the field. Unfortunately, many study findings only support previously demonstrated results, expanding them to a greater diversity of animals. Notably, some of the data are still at a very descriptive stage (based on observed correlations), and no evidence of adaptation mechanisms is presented. Finally, the initial part of the manuscript reduced my global appreciation of the manuscript without sufficient finding novelty. I hope my below comments will help to improve the manuscript.

Abstract:

The abstract does not reflect the importance of the results presented in the manuscript, and I would suggest the following modifications:

L6: provide a context to the sentence "Five major events of adaptation to the gut in the Archaea were identified." as the nature of those events is unclear without the main text.

L8-12: is it a surprising and novel result?

L11-13: this sentence is quite vague and does not provide accurate information

Introduction:

In the introduction, the questions, associated hypotheses and objectives of the study are not clearly presented. I am missing the evidence of the study's relevance in light of previous knowledge. Is the most significant aim of this study to expand the archaeome research to a more extensive range of animals than those analysed previously?

The final paragraph, which summarises the work performed in the study, is confusing as methane metabolism is advanced initially (L40), but non-methanogens are discussed in the following sentence (L42).

Results and discussion:

Archaea are present in the gut microbiome throughout the animal kingdom

- The first approach presented to quantify the microbial communities involve some specific qPCR targeting a few distinct groups. While methanogens are justified in the context of the study, analysing the Thaumarchaeota is surprising and lacks a strong justification.

- The universal sequencing approach would strongly depend on the sequencing depth and on the bias for several archaeal groups (including Thaumarchaeota). The results (figure 1 and associated text) indicate archaea's lower abundance (and diversity) using this methodology. As explained in L 64-68, an average of 10k reads per sample is likely insufficient for characterising the entire microbial community structure. Therefore, these results should not be included in the manuscript as they likely suffer from important biases.

- The discussion related to figure 1 is minimal, and one may wonder why the data are classified into 6 animal groups while results are not discussed in light of this classification.

Five major events of adaptation to the gut in the Archaea

- The events of gut adaptation presented in figure 2 lack clarity. The legend indicates that "orange arrows on the tree indicate proposed adaptations to the gut environment", suggesting 9 independent events. However, presence in the gut may not infer adaptation, so this representation is misleading. Importantly, no mechanism is explained for the adaptation.

The most crucial point of Figure 2 is the percentage of reads attributed to each taxon in the present study, including only gut-related samples. However, the data representation does not represent the extent of the findings. For example, it is mentioned (in L84) that 3 lineages are the "main methanogenic lineages in a much wider range of animals", but figure 2a only represents the high proportion of these lineages without associated distribution in a range of animals. The representation of the data in figure 3 is more informative, and I would suggest removing figure 2 from the main figures.

- This descriptive section is based on very short reads, so the high identity percentages reported here are not surprising (especially for the 16S rRNA marker gene). In addition, the inference made for figure 2d (based on the size of the fragments) is not highly convincing (in my opinion). It is justified to affiliate those reads to the Nitrososphaerales (and even Nitrosocosmicus) groups, but I would suggest restricting the similarity analysis at this level.

- Is it justifiable to use the silva database for the relative abundance of the reads in different environments? This may be the case, but the insertion of this justification will reinforce the present study results.

- The legend of figure 2c is unclear, especially the sentence "Lines indicate the moving averages with a subset size of 25 samples."

- The discussion associated with figure 2 (L 91-119) is too speculative, and the study presents some correlations without evidence of competitive or adaptative mechanisms.

Non-methanogenic lineages are components of the animal gut archaeome

- The bias of universal primers for Thaumarchaeota has been reported in previous studies, so the non-detection of this lineage using commonly used universal primers is not surprising.

- This section is based on logical observations but remains highly hypothetical. The sentence "The degree of adaptation and role of Nitrososphaerales in the gut remains to be elucidated." reflects well my current feeling of hypothetical discussion.

Specific associations between archaea and their hosts

- Figure 3 is beautiful and informative! I would have liked to see it much earlier in the manuscript to avoid any frustration build up. A minor improvement would be removing the sentence "Green represents animals with low richness, black represents medium richness ~ 25; and red represents high richness >50 archaeal ASVs." or adding similar sentences for panels a and b. Also, the origin of the animal phylogeny is not cited in the legend and panel e) is slightly unclear.

- Table 1 indicates some correlations between potential host factors and archaeal diversity. The "host order" is interpreted as host phylogeny, but a refined approach such as phylogenetic signal could be used here. In addition, while significant, the correlations are not strong ($R^2 = 0.1$ to 0.4). Clarification of the meaning of "coefficient of gut differentiation", especially in comparison to "digestive tract type" would be appreciated at this stage even if more details are given later in the manuscript.

- Figure 4 aims to demonstrate the existence of host specialisation for several archaeal lineages. Nonetheless, I am confused by this display as the highlighted squares are not explained, nor are each column of the heatmap. I supposed that the squares represent archaeal lineages widely present in some animal clades. However, the distribution of the "Insects and Insectivores clades" in a single individual of carnivorous (in red) or pholidota (in purple) does not correspond to this explanation.

- I would like to see the statistical results supporting all statements. For example, in L195- 199, "Archaea were also detected in a lower proportion of bird and fish species than in other animal classes (Figure 1; Figure S8). In Mammalia, samples belonging to Perissodactyla, Cetartiodactyla, Primates, Diprotodontia and Rodentia have the highest absolute concentration of archaea, whereas those belonging to Carnivora and Pholidota have the lowest (Figure 3b)." but both Figure 1 and Figure 3b represent data without any statistical analysis.

- The statistical comparison represented in Figure 5a is challenging to follow due to the numerous comparisons. As a rank was suggested, could letters be included to illustrate the ranking rather than all pairwise statistical differences? In addition, in the Wilcoxon rank sum test, I wonder if the difference in archaeal abundance can be concluded given the dissimilar data dispersion between groups? This test is quite robust to the heterogeneity of variance, although the 'shape' of the group (mammals, birds etc) distributions (gene copies/g) should be similar if comparing medians. If the 'shape' of the distribution is dissimilar between groups, then the test is only comparing distributions. The distributions are not similar based on the boxplots, so caution should be used when interpreting these results. Also, I understand the authors have performed a Kruskal Wallis test and then performed a large number of individual pairwise comparisons using Wilcoxon rank sum test. Due to the large number of tests performed, correcting for Type I error rate using a Bonferroni correction or similar should be performed to avoid increasing Type I error rate, but the correction the authors applied here (if any) was not indicated in the M&M.

In terms of the interpretation, it looks like the authors may be overstating some of these results with the sentence "mammals have the highest absolute abundance of archaea, followed by reptiles and amphibians" (in L191) as it looks to me that there is no difference between reptiles and mammals. Still, there is a difference between mammals and aves and between reptiles and aves.

Strong influence of diet on methanogen abundance and composition

- Figure 6d represents one of the study's major results, and I would advise highlighting this exciting finding. In contrast, figures 6a,b,c are confusing as a significant influence of diet is observed for the three groups (methanogens, Thaumarchaeota and bacteria), but only the first group is meaningful.
- In addition, the same statistical comments as for figure 5 applies to figure 6. The interpretation "the absolute and relative abundance of methanogens is higher in animals with a plant-based diet (e.g., leaves, fruits) than in animals feeding on meat or insects, and their abundance is intermediate in omnivorous animals (Figure 6a)" (L211) also seems to be an overstatement as animals with a fruit diet have the same abundance of methanogens than carnivorous or omnivorous animals for example.
- The beginning of sections L226 to 239 suggests that the present study's findings are not really novel, demonstrating high methanogen abundance in a more extensive range of animals than previously tested. As a non-specialist of methanogenesis, I would suggest shortening the section from L226 to 364 as this discussion is based on results that could be viewed as "cherry-picking" OTUs of interest.

Material and Methods

- A dilution following DNA quantification would be preferable to "random 10x dilution", especially for the qPCR assays. Indeed, insufficient dilution may lead to qPCR bias, and the absence of qPCR inhibition is essential in this study as many interpretations are based on these data. Based on the data provided in the supplementary table, the DNA concentrations ranged from 120 to 180202 ng per g of faeces, which represents very different concentrations. Some qPCR inhibition is likely to have happened for the more concentrated samples.
- The primer pairs used for archaeal qPCR were released in 1992 and 2004, which is anterior to the discovery of many archaeal clades. Does an in silico analysis confirm the specificity of these primers with these archaeal clades?
- Some statistical comments are indicated above

Other comments:

A careful revision of the presentation of the manuscript is required. A revision of the English is required (for example in L50: replace tree by 3). This also includes (but is not limited to) comma to be replaced by dots in numbers (for example in figures 2b or 2c) , missing bracket on the x axis in figure 2c or abbreviations used on figures (instead of full letter words, such as "arc-spec" in figure 1 or "methano" in figure 2) or legends (MiSeq). A thorough revision of the presentation would facilitate the reading of the manuscript.

Reviewer #2 (Remarks to the Author):

This is a most comprehensive study of abundance and diversity of archaea in more than 260 species of animals ranging from invertebrate to vertebrate hosts. The thorough analysis identified the factors that shape archaeal diversity and abundance in this habitat and extends the concept of the widespread nature of methanogens in the animal guts brought forward in the seminal study of Stumm and Hackstein more than 25 years ago (PNAS, 1994). The comprehensive scale of the study is unprecedented and provides a treasure trove of information for future investigations. In addition to methanogens, the study also covers the presence two of other archaeal lineages previously underrepresented in studies of intestinal habitats, i.e., Thaumarchaeota and Bathyarchaeota, in certain animals.

The authors comprehensively characterized archaeal community structure using a combination of three different approaches: quantitative PCR and amplicon sequencing of the 16S rRNA genes of archaea and of the entire community of prokaryotes. This provided not only qualitative information on archaeal diversity but also quantitative information on the abundance of each group. Most importantly, it revealed different ranges of detection for particular lineages and documented that universal primers often caused many archaeal groups in the animal gut to be entirely overlooked or underrepresent in their abundance.

The enormous dataset allowed not only to document the distribution of particular methanogenic lineages among animals but also to identify factors that influence their abundance, such as diet, digestive tract physiology, and host group. This provides new insights into the parallel evolution of methanogens in the gut microbiota of particular host groups. In addition, the co-occurrence analysis of bacteria and archaea revealed possible biotic interactions among the members of different domains.

The study is of great value for the field and will provide a solid foundation for future studies on the evolution and adaptation of archaea in the digestive tracts of animals. The analyses are state of the art and the manuscript has been carefully written and edited. It was a pleasure to read. I have only have a few minor comments:

48. Do not capitalize vernacular names (invertebrates, mammals).

50. 'three' approaches?

77-80. According to the microbiological code, ALL taxa with standing in nomenclature should be set in italics (i.e., also orders). Candidate taxa, however, should NOT be italicized (e.g., 'Ca. Methanomethylophilaceae'). By contrast, the zoological code only requires that only genus and species names (but not family names) are set in italics (line 259, Lemuridae). Illogical, but formally correct.

84. humans.

90. Methanimicrococcus occurs in all methanogenic insects orders (https://doi.org/10.1007/978-3-319-53114-4_13-1).

335-338. It may be of interest to point out that the Giant Panda is a herbivore that evolved from carnivorous ancestors, which would explain the relatively short retention time of the digesta.

393. better: 'Ca. Methanomethylophilus alvi' (see <https://doi.org/10.1099/ijsem.0.001715>; no italics)

.

Fig. 7b: I did struggle with this figure. It seems that the legend was cut off. The text (l. 289-298) didn't really help to explain the point the authors are trying to make. The methods did not show the underlying data but refer to the literature. In general, this could be an interesting discussion point, but the concept needs to be introduced better, and data used for the figure should be shown at least in the suppl. material.

REVIEWER COMMENTS

Reviewer #1 (Remarks to the Author):

This manuscript presents some interesting data on the characterisation of the archaeome in a variety of animals. Some of the findings on the link between archaeal diversity and host phylogeny (Fig 3) or correlation of methanogen abundance with diet-type over an extensive range of animals (Fig 6d) represent a substantial advance for the field. Unfortunately, many study findings only support previously demonstrated results, expanding them to a greater diversity of animals. Notably, some of the data are still at a very descriptive stage (based on observed correlations), and no evidence of adaptation mechanisms is presented. Finally, the initial part of the manuscript reduced my global appreciation of the manuscript without sufficient finding novelty. I hope my below comments will help to improve the manuscript.

>First of all, thank you for your detailed comments on the manuscript. We believe it helped improving it.

The first part of the manuscript is very important to us because it clarifies which archaea are present / dominant in the gut and allow to propose possible events of adaptation to the gut in the Archaea.

*We appreciate that R1 wants to know more about adaptation of archaea to the gut. Our study is based on 16S rRNA only, and it is not suited to tackle this aspect. We have recently published a paper on the adaptation of one lineage of archaea (*Methanimicrococcus*) to the gut based on genome sequences (Thomas et al., 2021; <https://doi.org/10.1038/s43705-021-00050-y>). Further analysis of the genes involved in adaptation at the scale of the archaeal domain will represent a future study on its own and will be guided by the results of the present study.*

Concerning novelty, to our knowledge, a global analysis of the phylogenetic distribution of host-associated and environmental archaea, as presented in figure 2a, has never been published before. In combination with our sequencing data, this allows to answer the question of which archaea are strongly associated to the gut and which ones are not.

Abstract:

The abstract does not reflect the importance of the results presented in the manuscript, and I would suggest the following modifications:

L6: provide a context to the sentence "Five major events of adaptation to the gut in the Archaea were identified." as the nature of those events is unclear without the main text.

L8-12: is it a surprising and novel result?

> It is a novel result. Increase in methanogen activity (in terms of gene expression or methane production) due to an increased diet fibre content was previously reported for human, pig, and ruminants, but this is the first time that fibre is linked to methanogen abundance in such a large range of hosts.

L11-13: this sentence is quite vague and does not provide accurate information

> We changed this sentence by "Our results provide key elements toward elucidation of the role of the archaeome in the gut environment, an emerging and important field of investigation."

Introduction:

In the introduction, the questions, associated hypotheses and objectives of the study are not clearly presented. I am missing the evidence of the study's relevance in light of previous knowledge. Is the most significant aim of this study to expand the archaeome research to a more extensive range of animals than those analysed previously? The final paragraph, which summarises the work performed in the study, is confusing as methane metabolism is advanced initially (L40), but non-methanogens are discussed in the following sentence (L42).

> We modified the introduction to better present the objectives and removed the reference to non-methanogenic archaea at this point.

Results and discussion:

Archaea are present in the gut microbiome throughout the animal kingdom

- The first approach presented to quantify the microbial communities involve some specific qPCR targeting a few distinct groups. While methanogens are justified in the context of the study, analysing the Thaumarchaeota is surprising and lacks a strong justification.

>The presence of Thaumarchaeota in animal samples has been reported in a few papers. This is now made clearer in the Introduction.

Also, before starting this work, we analysed 16S rRNA data from studies that were not focused on archaea and that used universal primer for sequencing, and we found the presence of Thaumarchaeota in several animal gut samples. In some of them they were even the dominant archaea. This is why we decided to include the Thaumarchaeota in our quantification analysis. This is now made clearer in the Introduction.

- The universal sequencing approach would strongly depend on the sequencing depth and on the bias for several archaeal groups, including Thaumarchaeota. The results (figure 1 and associated text) indicate archaea's lower abundance (and diversity) using this methodology. As explained in L 64-68, an average of 10k reads per sample is likely insufficient for characterising the entire microbial community structure. Therefore, these results should not be included in the manuscript as they likely suffer from important biases.

> Rarefaction at 10k read is similar to what is commonly in other recent studies on the gut microbiota (e.g. Boer et al., Nature Com 2019 (10k reads); Moeller et al., PNAS 2017 (10k reads); Song et al., 2020 mBio (10k reads)). The idea here is to compare how archaea are detected with different approaches using the same samples. This has never been done before at this scale. We think it is important to explain why the utilization of universal primers can lead to miss the detection of archaea in the samples.

- The discussion related to figure 1 is minimal, and one may wonder why the data are classified into 6 animal groups while results are not discussed in light of this classification.

> Samples were classified at the class level, except for Invertebrates for which there were very few species in each class. Gathering the samples at a lower host taxonomy level

(order) would have made the graph hard to read and they would have been almost all pooled at a higher taxonomic level. We refer to this level classification several times in the text and we cite this figure once after the first part, line 217 and we also use this level in figure 5.

Five major events of adaptation to the gut in the Archaea

- The events of gut adaptation presented in figure 2 lack clarity. The legend indicates that "orange arrows on the tree indicate proposed adaptations to the gut environment", suggesting 9 independent events.

>We have now clarified this by using large and small arrows to distinguish what we identified as the five main events of adaptation to the gut (leading to archaeal lineages dominantly present in the animal gut), from other events of adaptation to the gut discussed in the text. We also explain that in the new version of the figure legend.

We modified the title of the part for: "Five major lineages are dominant in the animal gut and suggest independent adaptations"

However, presence in the gut may not infer adaptation, so this representation is misleading. Importantly, no mechanism is explained for the adaptation.

> We agree that presence does not directly imply adaptation. However, the overrepresentation of a few lineages (present at high abundance) in the gut indicates they have traits allowing them to flourish in this environment (while most other archaea cannot). Their strong association with the gut environment vs other environments suggests that they specialized on the gut and cannot grow efficiently outside.

*Our study is based on 16S rRNA gene analyses, so we don't have the possibility to describe mechanisms here, but it allowed us to identify lineages in which these adaptations have most likely occurred. As said above, we have recently published a first study on the adaptation to the gut in one of these lineages of archaea (*Methanimicrococcus*)*

The most crucial point of Figure 2 is the percentage of reads attributed to each taxon in the present study, including only gut-related samples. However, the data representation does not represent the extent of the findings. For example, it is mentioned (in L84) that 3 lineages are the "main methanogenic lineages in a much wider range of animals", but figure 2a only represents the high proportion of these lineages without associated distribution in a range of animals. The representation of the data in figure 3 is more informative, and I would suggest removing figure 2 from the main figures.

>The goal of Figure 2 is to provide an overview of host-associated archaea before considering the factors that influence their diversity and abundance. We summarize below why we think this figure is important for the paper.

We integrated our data with data from a wide range of studies (from Silva) to identify lineages that are dominant in the gut (using our data) and almost only present in this environment (using Silva data; Fig 2A). Even if the Silva data are not from us, their analysis in terms of environmental distribution (gut/environment/engineered) is new (to our knowledge) and useful to draw our conclusions. Because the identification of a large number of Thaumarchaeota sequences in our study contrasts with most previous studies (Fig. 2A), we looked in more detail at their abundance and show that they are widespread

but generally occur at low absolute abundance, and they dominate the gut archaeome when methanogens have a low absolute abundance (Fig. 2B and 2C). In the frame of this focus on Thaumarchaeota we also present a tree showing that the most widespread Thaumarchaeota in the gut of animal are actually closely related or similar to the dominant archaea in soil (Fig. 2D). We think it is an important point to raise and discuss. Finally, figure 2B also shows that even if the rarest archaeal taxa can represent the majority of archaea in some animal gut, they always occur in low absolute abundance (Fig. 2B).

We removed the sentence L84 as the aspects related to the distribution of archaeal lineages among host is presented after.

- This descriptive section is based on very short reads, so the high identity percentages reported here are not surprising (especially for the 16S rRNA marker gene). In addition, the inference made for figure 2d (based on the size of the fragments) is not highly convincing (in my opinion). It is justified to affiliate those reads to the Nitrososphaerales (and even Nitrosocosmicus) groups, but I would suggest restricting the similarity analysis at this level.

> *We believe that a tree is more talkative and integrative than similarity analysis alone (and we also provide similarity values).*

- Is it justifiable to use the silva database for the relative abundance of the reads in different environments? This may be the case, but the insertion of this justification will reinforce the present study results.

> *The 16S rRNA gene sequences from the Silva database originate from hundreds of studies covering all types of environments and have a robust taxonomic annotation integrating most of the phylogenetic diversity. It is thus a great tool to define the distribution of archaeal lineages in all these environments. We have added a sentence in the Material and Method section concerning this aspect.*

- The legend of figure 2c is unclear, especially the sentence "Lines indicate the moving averages with a subset size of 25 samples."

> *We changed the sentence for "Coloured lines indicate the moving averages of the relative abundance of these three groups on a subset size of 25 samples".*

- The discussion associated with figure 2 (L 91-119) is too speculative, and the study presents some correlations without evidence of competitive or adaptative mechanisms.

> *We answered to this point above. The adaptation events that we propose here (based on our data, data from a wide range of environments, relative and absolute abundance of gut archaea) are the base for future comparative genomics and experimental studies.*

Non-methanogenic lineages are components of the animal gut archaeome

- The bias of universal primers for Thaumarchaeota has been reported in previous studies, so the non-detection of this lineage using commonly used universal primers is not surprising.

> *There is no mismatch between the universal primers we used and the Thaumarchaeota, so we don't see why there should be a PCR bias. The study of Bates et al., 2010, which identified the dominant archaea (i.e. Thaumarchaeota) in soil samples relied on universal prokaryotic primers. Rather than a PCR bias, our results suggest that Thaumarchaeota are not detected in the gut because they almost always occur at low (relative) abundance in this environment (Figure 2b and 2c).*

Of course, a bias is possible in other studies depending on the universal primer set that is used.

- This section is based on logical observations but remains highly hypothetical. The sentence "The degree of adaptation and role of Nitrososphaeraceae in the gut remains to be elucidated." reflects well my current feeling of hypothetical discussion.

> *The report of Thaumarchaeota being the dominant gut archaea in several hosts is somehow unique to this study. So, we feel it is necessary to discuss this observation and to raise that the dominant soil archaea also correspond to the most widespread gut Thaumarchaeota. We just bring forward information which could be of interest for future investigations.*

Specific associations between archaea and their hosts

- Figure 3 is beautiful and informative! I would have liked to see it much earlier in the manuscript to avoid any frustration build up. A minor improvement would be removing the sentence "Green represents animals with low richness, black represents medium richness ~ 25; and red represents high richness >50 archaeal ASVs." or adding similar sentences for panels a and b. Also, the origin of the animal phylogeny is not cited in the legend and panel e) is slightly unclear.

> *Thank you, we removed the sentence as suggested, added the origin of the animal tree and clarified the legend of panel e.*

- Table 1 indicates some correlations between potential host factors and archaeal diversity. The "host order" is interpreted as host phylogeny, but a refined approach such as phylogenetic signal could be used here. In addition, while significant, the correlations are not strong ($R^2 = 0.1$ to 0.4). Clarification of the meaning of "coefficient of gut differentiation", especially in comparison to "digestive tract type" would be appreciated at this stage even if more details are given later in the manuscript.

> *Thank you for these suggestions. Using the Mantel test we found that the mammal phylogeny is a major factor influencing archaeal community structure. The results from this test have been added to Table 1. Further, we have clarified the meaning of coefficient of gut differentiation and digestive tract as suggested.*

- Figure 4 aims to demonstrate the existence of host specialisation for several archaeal lineages. Nonetheless, I am confused by this display as the highlighted squares are not explained, nor are each column of the heatmap. I supposed that the squares represent archaeal lineages widely present in some animal clades. However, the distribution of the "Insects and Insectivores clades" in a single individual of carnivorous (in red) or pholidota (in purple) does not correspond to this explanation.

> Thank you helping clarifying this. We added explanations on these aspects in the revised legend of Figure 4: "Black boxes highlight archaeal clades preferentially present in a given host order, except for the Insect+insectivores clade composed of archaea preferentially present in insectivorous animals from different mammalian orders and insects (mostly termites, sequences from the literature)."

- I would like to see the statistical results supporting all statements. For example, in L195-199, "Archaea were also detected in a lower proportion of bird and fish species than in other animal classes (Figure 1; Figure S8).

> We now provide statistics for all statements, and modified this sentence.

- In Mammalia, samples belonging to Perissodactyla, Cetartiodactyla, Primates, Diprotodontia and Rodentia have the highest absolute concentration of archaea, whereas those belonging to Carnivora and Pholidota have the lowest (Figure 3b)." but both Figure 1 and Figure 3b represent data without any statistical analysis.

> We changed the reference to figure 3 for a reference to figure S10 that displays results of statistical analysis.

- The statistical comparison represented in Figure 5a is challenging to follow due to the numerous comparisons. As a rank was suggested, could letters be included to illustrate the ranking rather than all pairwise statistical differences? In addition, in the Wilcoxon rank sum test, I wonder if the difference in archaeal abundance can be concluded given the dissimilar data dispersion between groups? This test is quite robust to the heterogeneity of variance, although the 'shape' of the group (mammals, birds etc) distributions (gene copies/g) should be similar if comparing medians. If the 'shape' of the distribution is dissimilar between groups, then the test is only comparing distributions. The distributions are not similar based on the boxplots, so caution should be used when interpreting these results. Also, I understand the authors have performed a Kruskal Wallis test and then performed a large number of individual pairwise comparisons using Wilcoxon rank sum test. Due to the large number of tests performed, correcting for Type I error rate using a Bonferroni correction or similar should be performed to avoid increasing Type I error rate, but the correction the authors applied here (if any) was not indicated in the M&M.

> Thank you for these suggestions. Figure 5 has now been changed in an effort to make the pairwise differences easier to identify. Additionally, the text has been changed to better describe the differences between animal classes and the abundances of archaea and bacteria.

The Benjamini & Hochberg correction was used in our analyses, however we did not clearly indicate this in the methods. This has been corrected.

In terms of the interpretation, it looks like the authors may be overstating some of these results with the sentence "mammals have the highest absolute abundance of archaea, followed by reptiles and amphibians" (in L191) as it looks to me that there is no difference between reptiles and mammals. Still, there is a difference between mammals and aves and between reptiles and aves.

> *Thank you for identifying this. We modified the sentence to state that “mammals and reptiles tend to host higher abundances of archaea, than other animal groups like amphibians, birds, fish, and invertebrates (Figure 5a)”*

Strong influence of diet on methanogen abundance and composition

- Figure 6d represents one of the study's major results, and I would advise highlighting this exciting finding. In contrast, figures 6a,b,c are confusing as a significant influence of diet is observed for the three groups (methanogens, Thaumarchaeota and bacteria), but only the first group is meaningful.

> *Thank you for this suggestion, we have changed Figure 6 to highlight the influence of diet type and fibre consumption on methanogens by moving panels concerning bacteria and Thaumarchaeota to a new supplementary figure 12.*

- In addition, the same statistical comments as for figure 5 applies to figure 6. The interpretation "the absolute and relative abundance of methanogens is higher in animals with a plant-based diet (e.g., leaves, fruits) than in animals feeding on meat or insects, and their abundance is intermediate in omnivorous animals (Figure 6a)" (L211) also seems to be an overstatement as animals with a fruit diet have the same abundance of methanogens than carnivorous or omnivorous animals for example.

> *We changed the sentence lines 216-218. “Moreover, the absolute and relative abundance of methanogens is higher in animals feeding on leaves than in animals feeding on meat or insects, and it tends (not statistically supported) to be intermediate in omnivorous animals (Figure 6a).”*

- The beginning of sections L226 to 239 suggests that the present study's findings are not really novel, demonstrating high methanogen abundance in a more extensive range of animals than previously tested.

> *We started the paragraph from data present in the literature to compare with what we found as it allowed a better transition to the role of specific bacteria in H₂ production. There are a few data known from human. Here, indeed, we support these observations with an extensive range of animals.*

- As a non-specialist of methanogenesis, I would suggest shortening the section from L226 to 364 as this discussion is based on results that could be viewed as "cherry-picking" OTUs of interest.

> *We are not only discussing OTUs and co-occurrence in this section but also the dominant types of methane metabolisms in relation to diet, and the effect of gut physiology on gut archaea.*

Material and Methods

- A dilution following DNA quantification would be preferable to "random 10x dilution", especially for the qPCR assays. Indeed, insufficient dilution may lead to qPCR bias, and the

absence of qPCR inhibition is essential in this study as many interpretations are based on these data. Based on the data provided in the supplementary table, the DNA concentrations ranged from 120 to 180202 ng per g of faeces, which represents very different concentrations. Some qPCR inhibition is likely to have happened for the more concentrated samples.

> For most samples, there is a good correlation between the concentration of extracted DNA and the bacterial 16S rRNA gene copy number and there is no observed plateau in the 16S rRNA gene copy number for the highest DNA concentration (new figure S19). Five to six samples with a high DNA concentration (above 100000 ng of DNA /g of feces; i.e. 1.8 ng/ul of PCR reaction) have a low 16S rRNA gene copy number which might reflect an inhibition. However, if there is a bias due to DNA concentration it had overall a minor effect on the results.

By checking this aspect raised by reviewer 1, we found that a number of samples have a low 16S rRNA gene copy number compared to their DNA concentration (Figure S19). These samples mainly correspond to Aves and Actinopterygii (but not all Aves and Actinopterygii samples). As this low ratio may reflect an inhibition of the qPCR assay we decided to remove these samples from the analysis of qPCR data (Figure S19). Their removal did not affect the main results reported in this study, especially because many analyses focused on Mammal species. It mainly changes the results presented for archaea (Figure S9) and bacteria (Figure 5b, Figure 12b), with a lower number of significant differences in abundance between categories (host class and host diets). On Figure 3 and Figure S8 we added stars on the abundance heat map to indicate that abundance of archaea in these species might be underestimated.

Overall, we think that these modifications minimize the risks of presenting results possibly biased by qPCR inhibition. But again, it didn't strongly affect the main results.

- The primer pairs used for archaeal qPCR were released in 1992 and 2004, which is anterior to the discovery of many archaeal clades. Does an in-silico analysis confirm the specificity of these primers with these archaeal clades?

> One of the two is from actually from 2011. Yes, an in silico analysis was performed, they target 77% of the archaea with no mismatch and 91% with one mismatch. These primers were selected among many, not only to get the largest number of archaea, but also to avoid amplifying bacteria. The abundance of total archaea is strongly correlated with the sum of all methanogens (fig 2c), and even better correlated when Thaumarchaeota are added (not shown). So it's likely that we didn't miss a lot of the total archaea.

- Some statistical comments are indicated above

Other comments:

A careful revision of the presentation of the manuscript is required. A revision of the English is required (for example in L50: replace tree by 3). This also includes (but is not limited to) comma to be replaced by dots in numbers (for example in figures 2b or 2c) , missing bracket on the x axis in figure 2c or abbreviations used on figures (instead of full letter words, such as "arc-spec" in figure 1 or "methano" in figure 2) or legends (MiSeq). A thorough revision of the presentation would facilitate the reading of the manuscript.

> *Thank you for reporting these typos. We corrected them as well as others to improve the reading of the manuscript.*

Reviewer #2 (Remarks to the Author):

This is a most comprehensive study of abundance and diversity of archaea in more than 260 species of animals ranging from invertebrate to vertebrate hosts. The thorough analysis identified the factors that shape archaeal diversity and abundance in this habitat and extends the concept of the widespread nature of methanogens in the animal guts brought forward in the seminal study of Stumm and Hackstein more than 25 years ago (PNAS, 1994). The comprehensive scale of the study is unprecedented and provides a treasure trove of information for future investigations. In addition to methanogens, the study also covers the presence two of other archaeal lineages previously underrepresented in studies of intestinal habitats, i.e., Thaumarchaeota and Bathyarchaeota, in certain animals.

The authors comprehensively characterized archaeal community structure using a combination of three different approaches: quantitative PCR and amplicon sequencing of the 16S rRNA genes of archaea and of the entire community of prokaryotes. This provided not only qualitative information on archaeal diversity but also quantitative information on the abundance of each group. Most importantly, it revealed different ranges of detection for particular lineages and documented that universal primers often caused many archaeal groups in the animal gut to be entirely overlooked or underrepresented in their abundance.

The enormous dataset allowed not only to document the distribution of particular methanogenic lineages among animals but also to identify factors that influence their abundance, such as diet, digestive tract physiology, and host group. This provides new insights into the parallel evolution of methanogens in the gut microbiota of particular host groups. In addition, the co-occurrence analysis of bacteria and archaea revealed possible biotic interactions among the members of different domains.

The study is of great value for the field and will provide a solid foundation for future studies on the evolution and adaptation of archaea in the digestive tracts of animals. The analyses are state of the art and the manuscript has been carefully written and edited. It was a pleasure to read. I have only have a few minor comments:

> *Thank you very much for your support and comments.*

48. Do not capitalize vernacular names (invertebrates, mammals).

>*Done.*

50. 'three' approaches?

>*Done.*

77-80. According to the microbiological code, ALL taxa with standing in nomenclature should be set in italics (i.e., also orders). Candidate taxa, however, should NOT be italicized (e.g., 'Ca. Methanomethylophilaceae'). By contrast, the zoological code only requires that

only genus and species names (but not family names) are set in italics (line 259, Lemuridae). Illogical, but formally correct.

>This was modified.

84. humans.

>Done.

90. Methanimicrococcus occurs in all methanogenic insects orders (https://doi.org/10.1007/978-3-319-53114-4_13-1).

>We added this reference to the text.

335-338. It may be of interest to point out that the Giant Panda is a herbivore that evolved from carnivorous ancestors, which would explain the relatively short retention time of the digesta.

>Thank you, we added this precision.

393. better: ‘Ca. Methanomethylophilus alvi’ (see <https://doi.org/10.1099/ijsem.0.001715>; no italics) .

>Thank you for the suggestion, however we prefer to keep M. alvus here as it is at present the most commonly used named for this species.

Fig. 7b: I did struggle with this figure. It seems that the legend was cut off. The text (l. 289-298) didn’t really help to explain the point the authors are trying to make. The methods did not show the underlying data but refer to the literature. In general, this could be an interesting discussion point, but the concept needs to be introduced better, and data used for the figure should be shown at least in the suppl. material.

>Thanks for the comment, some details were indeed lacking and were added to the text, figure and in a new supplementary table (Table S5). We hope that it is clearer now.

REVIEWERS' COMMENTS

Reviewer #1 (Remarks to the Author):

The authors have made a substantial effort to improve their manuscript and clarify some of the issues I previously have. I found the results much clearer and appreciate the efforts made by the authors to address my comments. I only have a few remaining minor comments:

The title change “Five major lineages are dominant in the animal gut and suggest independent adaptations” could be modified to “Dominance of five major unrelated lineages in the animal gut suggest independent adaptations”

L160: I suggest removal of the sentence “Moreover, while Thaumarchaeota group I.1c are among the dominant archaea in soil 39,49, we only found two ASVs belonging to this lineage in one sample.” as they are mainly present in acidic soils, which probably does not reflect the pH of the gut (per the technique used in this study).

L166: change to “can be maintained”

L192. Change “this is mainly” to “This is the case”

Finally, I could still spot a few typos but hopefully they will be detected at the final stage

Reviewer #2 (Remarks to the Author):

In my review of the original submission, I had already indicated that I found the study is of great value for the field and will provide a solid foundation for future studies on the evolution and adaptation of archaea in the digestive tracts of animals.

My few (minor) comments have been addressed to my satisfaction.

For the sake of the argument:

I am still not convinced that the concept behind Fig. 7B is properly explained. I consider this a only minor problem and leave it up to the authors to address. It becomes clear what the authors mean to say, but the logic of the arguments for the "favorability" of a pathway remain obscure. First, the

overall thermodynamics of a reaction is independent of the underlying biochemical pathway. Therefore, the catabolism should be unambiguously defined by an equation! Second, the criterion deciding whether a pathway is more favorable than others should be explained. How is it possible that sometimes all catabolisms are most favorable?

There are several typos in the figure: "Possibly", hyphenation of "Methano-sphaera", missing spaces in legend.

RESPONSE TO REVIEWERS' COMMENTS

Reviewer #1 (Remarks to the Author):

The authors have made a substantial effort to improve their manuscript and clarify some of the issues I previously have. I found the results much clearer and appreciate the efforts made by the authors to address my comments.

I only have a few remaining minor comments:

The title change “Five major lineages are dominant in the animal gut and suggest independent adaptations” could be modified to “Dominance of five major unrelated lineages in the animal gut suggest independent adaptations”

> We changed the title accordingly to reviewer’s suggestion

L160: I suggest removal of the sentence “Moreover, while Thaumarchaeota group I.1c are among the dominant archaea in soil 39,49, we only found two ASVs belonging to this lineage in one sample.” as they are mainly present in acidic soils, which probably does not reflect the pH of the gut (per the technique used in this study).

>We think the reviewer misunderstood what we meant. We are not saying that Thaumarchaeota group I.1c should develop in the gut, but that if the Thaumarchaeota we detected are ingested by chance and are only transient, Thaumarchaeota group I.1c is expected to be more commonly detected in our samples. We modified the sentence to make it clearer. *“Moreover, while Thaumarchaeota group I.1c are among the dominant archaea in soil ^{44,47} and thus should also be commonly present in our samples if detected Thaumarchaeota are only ingested by chance, we only identified two ASVs belonging to this lineage in one sample.”*

L166: change to “can be maintained”

L192. Change “this is mainly” to “This is the case”

Finally, I could still spot a few typos but hopefully they will be detected at the final stage

> We corrected these typos and other, in the text and figure legends.

Reviewer #2 (Remarks to the Author):

In my review of the original submission, I had already indicated that I found the study is of great value for the field and will provide a solid foundation for future studies on the evolution and adaptation of archaea in the digestive tracts of animals.

My few (minor) comments have been addressed to my satisfaction.

For the sake of the argument:

I am still not convinced that the concept behind Fig. 7B is properly explained. I consider this

a only minor problem and leave it up to the authors to address. It becomes clear what the authors mean to say, but the logic of the arguments for the "favorability" of a pathway remain obscure. First, the overall thermodynamics of a reaction is independent of the underlying biochemical pathway.

> We agree that "favorability" was not the best term. Instead of saying that a pathway is more favorable others, we say that it yields more energy (under specific conditions). We corrected this in the text and the legend of the figure 7b.

Therefore, the catabolism should be unambiguously defined by an equation!

> We provide those equations in the method section.

Second, the criterion deciding whether a pathway is more favorable than others should be explained. How is it possible that sometimes all catabolisms are most favorable?

> We removed the term of "favorable". Now we explain that depending on the methanol concentration and H₂ partial pressure, pathway(s) yield more energy (they have a more negative ΔG) than other(s). We agree with the reviewer that the three pathways cannot be all the most favorable/have the highest energy yield (i.e. in the right panel legend of figure 7b). So we replaced "All three pathways" by "None*" (for conditions under which all three pathways yield the same amount of energy per mol of methane produced). We put a star to explain it in the figure legend.

There are several typos in the figure: "Possibly", hyphenation of "Methano-sphaera", missing spaces in legend.

> We corrected these typos and other, in the text and figure legends.